



# Two databases derived from BGC-Argo float measurements for biogeochemical and bio-optical applications at the global scale

Emanuele Organelli[1,2], Marie Barbieux[1], Hervé Claustre[1], Catherine Schmechtig[3], Antoine Poteau[1], Annick Bricaud[1], Emmanuel Boss[4], Nathan Briggs[1], Giorgio Dall'Olmo[2,5], Fabrizio D'Ortenzio[1], Edouard Leymarie[1], Antoine Mangin[6], Grigor Obolensky[7], Christophe Penkerc'h[1], Louis Prieur[1], Collin Roesler[8], Romain Serra[6], Julia Uitz[1], Xiaogang Xing[9]

[1] Sorbonne Universités, UPMC Univ Paris 06, CNRS, UMR 7093, Laboratoire d'Océanographie de Villefranche (LOV), 181 Chemin du Lazaret, 06230 Villefranche-sur-mer, France
[2] Plymouth Marine Laboratory, PL1 3DH Plymouth, United Kingdom
[3] Sorbonne Universités, UPMC Université Paris 06, CNRS, UMS 3455, OSU Ecce-Terra, Paris, France
[4] School of Marine Sciences, University of Maine, Orono, Maine, USA
[5] National Centre for Earth Observation, Plymouth Marine Laboratory, PL1 3DH Plymouth, United Kingdom
[6] ACRI-ST, 260 route du Pin Montard, 06904 Sophia Antipolis, France
[7] ERIC Euro-Argo, F-29280 Plouzané, France
[8] Department of Earth and Oceanographic Science, Bowdoin College, Brunswick, Maine, USA
[9] Second Institute of Oceanography, State Oceanic Administration, Hangzhou, 310012, China

*Correspondence to*: Emanuele Organelli (emo@pml.ac.uk)

**Abstract.** Since 2012, an array of 105 Biogeochemical (BGC) Argo floats has been deployed across the world's oceans to fill the observational gap characterizing most of open-ocean environments. Profiles of biogeochemical (chlorophyll and fluorescent dissolved organic matter) and optical (single-wavelength particulate optical backscattering, downward irradiance at three wavelengths and photosynthetically available radiation) variables are collected in the upper 1000 m every 1 to 10 days. The global database of 9837 vertical profiles collected up to January 2016 is presented and its spatial and temporal coverage is discussed. Each variable is quality controlled with specifically-developed procedures and its time-series is quality-assessed to identify issues related to biofouling and/or instrumental drift. A second database of 5748 profile-derived products within the first optical depth (i.e. the layer of interest for satellite remote sensing) is also presented and its spatio-temporal distribution discussed. This database, devoted to field and remote ocean color applications, includes diffuse attenuation coefficients for downward irradiance at three narrow wavebands and one broad waveband (photosynthetically available radiation), calibrated chlorophyll and dissolved organic matter fluorescence, and single-wavelength particulate optical backscattering. To demonstrate the applicability of these global databases, data within the first optical depth are finally compared with previously established bio-optical models and used to validate remotely-derived bio-optical products. The quality-controlled databases are publicly available from SEANOE (SEA scieNtific Open data Edition) publisher at http://doi.org/10.17882/49388 and http://doi.org/10.17882/47142 for vertical profiles and products within the first optical depth, respectively.





## 1 Introduction

In the early 2000s, the international oceanographic community raised concerns about the large uncertainties still limiting the estimation of key biogeochemical processes in the ocean that contribute to controlling the Earth's climate (e.g. primary production and carbon export). The spatial and temporal under-sampling of most of the world's oceans was considered the

main cause of this limitation (Munk, 2000; Hall et al., 2010). The same community thus proposed the implementation of autonomous platforms, such as the Biogeochemical-Argo profiling floats (hereafter BGC-Argo floats), as one solution to fill this observational gap (Johnson et al., 2009; Claustre et al., 2010a). Unlike sampling from vessels, observations by BGC-Argo floats operate with high temporal and spatial coverage, including unexplored and remote areas, and during periods when ship-based sampling is difficult. The BGC-Argo sampling approach can therefore help the scientific community

accumulate observations on biological and biogeochemical properties of the ocean, and characterize their natural variability from the surface to the ocean's interior in a new and systematic way (Claustre et al., 2010a; Biogeochemical-Argo Planning Group, 2016; Johnson and Claustre, 2016).

In 2012, an array of BGC-Argo floats started to be deployed in several oceanic areas encompassing a wide range of biogeochemical status and trophic conditions, from subpolar to tropical and from eutrophic systems to oligotrophic mid-

ocean gyres (Organelli et al., 2016a; 2017). This array of floats was devoted to the acquisition of profiles of key biogeochemical quantities via their optical properties (i.e. chlorophyll $a$, colored dissolved organic matter, nitrate concentrations) and of hydrological variables (i.e. temperature and salinity). In addition, the array provided measurements of the underwater light field (i.e. irradiance) and of the inherent optical properties (i.e. particulate optical beam attenuation and backscattering coefficients) of the oceans. All these measurements, and derived quantities, are useful both for

biogeochemical and bio-optical studies, to address the variability of biological processes (e.g. phytoplankton blooms and phenology; Lacour et al., 2015), to derive other biogeochemical quantities or fluxes (e.g. particulate organic carbon export, Dall'Olmo and Mork, 2014), and to validate bio-optical products retrieved from ocean color satellites (Claustre et al., 2010b; IOCCG, 2011, 2015; Gerbi et al., 2016).

The study reported here presents a quality-controlled database of biogeochemical and bio-optical vertical profiles acquired

by more than 100 BGC-Argo floats equipped with a homogeneous and inter-operable instrumental configuration. The "Biogeochemical and OPtical Argo Database – profiles" (BOPAD-prof; Barbieux et al., 2017) includes 0-1000 m measurements of calibrated chlorophyll $a$ (Chl, mg m$^{-3}$) and dissolved organic matter fluorescence (FDOM, ppb of quinine sulphate) concentrations, the particulate optical backscattering coefficient at 700 nm ($b_{bp}(700)$, m$^{-1}$), downward irradiance $E_d(\lambda)$ at three wavelengths (i.e., 380, 412 and 490 nm, µW cm$^{-2}$ nm$^{-1}$) and the spectrally-integrated photosynthetically

available radiation (PAR, µmol quanta m$^{-2}$ s$^{-1}$). Temperature (T, $^\circ$C) and salinity (S, psu) provide the hydrographic context for the optical observations. The geographic and temporal distribution of each parameter is described and discussed. A second database is specifically devoted to field and remote ocean-color applications (Organelli et al., 2016b). It is focused on observations and derived products within the first optical depth $Z_{pd}$ (also known as the penetration depth, i.e. the layer of



interest for satellite remote sensing; Gordon and McCluney, 1975; units of m), and includes the "Biogeochemical and OPtical Argo Database – surface" (BOPAD-surf) Chl, FDOM and $b_{bp}(700)$ quantities derived from the quality-controlled vertical profiles in addition to the diffuse attenuation coefficients for downward irradiance ($K_d(\lambda)$, m$^{-1}$) and PAR ($K_d$(PAR), m$^{-1}$). Finally, data presented in BOPAD-surf are compared with existing bio-optical models and used in conjunction with

products derived from satellite platforms in order to show applicability for validating ocean-color bio-optical products at the global scale.

## 2 Material and methods

### 2.1 Biogeochemical Argo floats: instrument, sampling strategy and data

The "PROVOR-CTS 4" profiling float used in this study is one of the latest models of autonomous platforms developed by

NKE Marine Electronics Inc. (France). Designed in the context of the Remotely-Sensed Biogeochemical Cycles in the Ocean (remOcean) and Novel Argo Ocean Observing System (NAOS) projects, this profiling float has been also adopted by several international collaborators and research programs. Technical description of platform and instrument arrangement can be fully found in Leymarie et al. (2013) and Organelli et al. (2016a).

All   "PROVOR-CTS 4" profiling floats were programmed to acquire 0-1000 m vertical profiles every 1 to 10 days depending

on mission and scientific objectives. Upward profiles commence from the 1000 m parking-depth in time for surfacing around local noon. Data acquisition was nominally 0.20 m resolution between surface and 10 m, 1 m resolution between 10 and 250 m, and generally 10 m resolution between 250 and 1000 m (except in some occasions where it was 1 m).

An array of 105 BGC-Argo floats acquired more than 10,000 vertical profiles of bio-optical and biogeochemical variables over a broad range of oceanic environments and trophic conditions between October 2012 and January 2016. A WETLabs

ECO (Environmental Characterization Optics) sensor installed on each BGC-Argo float provided 0-1000 m vertical profiles of chlorophyll (excitation/emission 470/695 nm) and dissolved organic matter (excitation/emission 370/460 nm) fluorescence, and of the volume scattering coefficient ($\beta(\theta, \lambda)$) measured at an angle of 124° and a wavelength $\lambda$ of 700 nm (Sullivan et al., 2013; Schmechtig et al., 2016). The multispectral ocean-color radiometer OCR-504 (SATLANTIC Inc.) provided 0-250 m vertical profiles of photosynthetically available radiation (PAR) and downward irradiance $E_d(\lambda)$ at three

wavelengths. Electronic counts of each measured variable were converted into geophysical quantities using calibration factors and practices provided by manufacturers (SATLANTIC, 2013; WETLabs, 2016). According to the standard procedures for Argo data management (Wong et al., 2015), each profile was then quality-controlled applying methods specifically developed for each parameter (see Sect. 2.2). In addition, because sensor performance might degrade over the float lifetime, time-series of each raw variable for each float were also evaluated for possible corruption by biofouling or

instrumental drift (see Sect. 2.3). Hence, a total of 9837 BGC-Argo stations, each one corresponding to an upward profile, composed the database BOPAD-prof presented in this study (Fig. 1). To discuss the geographic and temporal



representativeness of the database, the 9837 quality-controlled stations were grouped into 25 geographic areas (Table S1). The numbers, names and details of the included BGC-Argo floats are also shown in Table S1.

**2.2 Quality-control of vertical profiles**

Vertical profiles of chlorophyll *a* concentration (Chl) were quality-controlled following procedures and recommendations in

Schmechtig et al. (2014). Profiles were: 1) adjusted for non-zero deep values; 2) corrected by removing negative spikes lower than twice the 10-quantiles of the residual signal calculated as the difference between the profile values and a median filter (5 point window); 3) checked that measured values were within the specific range reported in the manufacturer's technical specifications (WETLabs, 2016); and 4) corrected for non-photochemical quenching (Kiefer, 1973) according to Xing et al. (2012). Profiles collected in areas such as the Black Sea and subtropical gyres were further corrected for the

contribution of fluorescence originating from non-algal matter following procedures described in Xing et al. (2017). The magnitude of the correction within the mixed layer depth varied between 3 and 50 % (see for details Table 2 in Xing et al., 2017 for the same database). Finally, values <0.014 mg m$^{-3}$ (i.e. ~2 digital counts measured by the fluorometer) were turned into NaN and, according to the recommendations by Roesler et al. (2017) on the overestimation by standard WETLabs fluorometers, remaining chlorophyll *a* values were divided by a factor of 2 to correct for the global bias in factory

calibration.

Vertical profiles of fluorescent dissolved organic matter concentration (FDOM) were quality-controlled according to the following procedures: 1) checked that measured values were within the specific range reported in the manufacturer's technical specifications (WETLabs, 2016); 2) removal of spikes outside the 25- and 75-quantiles of the raw profile, and then corresponding to measurements with an absolute value of the residual signal (calculated as the difference between the profile

values and a mean filter) > 4. Finally, according to the assumption that deep CDOM concentration is conservative in a given water body (Nelson et al., 2010), and considering the fact that BGC-Argo floats included in this database spent their lifetime mainly within the same deep water mass, an offset was applied to each FDOM profile to align the median value between 950 and 1000 m with the first profile and correct for possible sensor's drift.

Following procedures described in Schmechtig et al. (2016), vertical profiles of the angular scattering coefficient β(124°,

700) were: 1) converted into the particulate angular scattering coefficient by removing the contribution of pure seawater, which depends on water temperature and salinity (Zhang et al., 2009); 2) converted to the particulate optical backscattering coefficient at 700 nm ($b_{bp}$(700)) following procedures in Boss and Pegau (2001) and Sullivan et al. (2013); 3) verified for measured values according to the manufacturer's technical specifications (WETLabs, 2016); 4) corrected by removing negative spikes lower than twice the 10-quantiles of the residual signal calculated as the difference between the profile and a

median filter (5 point window).

Vertical profiles of PAR and $E_d(\lambda)$ were quality controlled following the procedures detailed in Organelli et al. (2016a). A first step of the quality control consisted in identifying and discarding each profile acquired under variable cloud and wave conditions (see quantitative metrics in Organelli et al., 2016a). Remaining profiles were quality controlled for identification



and removal of: 1) nonzero dark measurements at depth; 2) sporadic atmospheric clouds; and 3) wave focusing (Zaneveld et al., 2001) in the upper part of the profile. Finally, $E_d(0^-)$ and PAR just below the sea surface were extrapolated from the quality-controlled profile according to the methodology described in Organelli et al. (2016a).

Profile-by-profile analysis/visualization of each variable and related statistics are available on http://seasiderendezvous.eu.

**2.3 Testing for biofouling and instrumental drift**

Testing of sensor performances for potential biofouling and instrumental drift was conducted on raw time-series of salinity, chlorophyll $a$, FDOM, $b_{bp}(700)$ and $E_d(\lambda)$ collected by each of the 105 BGC-Argo floats. Each variable was examined both individually and in conjunction with the others, which is greatly aided by redundancy amongst derived quantities. The tests used to identify biofouling and instrumental drift were: 1) Comparison of measured $E_d(0^+)$ values with those estimated by the

Gregg and Carder (1990) model for irradiances with clear cloudless sky (Organelli et al., 2016a); 2) Analysis of the sensor's dark measurements at the 1000 m parking-depth over time (only for Chl and $E_d(\lambda)$); 3) identification of sharp gradients in measured variables over the entire profile (i.e. decrease of Chl and FDOM concentrations or increase in $b_{bp}(700)$ values) not attributable to any biological or hydrological cause (e.g. nepheloid layer of particles); and 4) analysis of the relationship between raw FDOM and salinity at the 1000 m parking-depth over time. Assuming that deep CDOM concentrations are

conservative in the same water body (Nelson et al., 2010), variability in deep FDOM and the presence of constant salinity is likely due to changes in sensor performances (Fig. 2). When the results of the tests above indicated possible measurement issues, each variable time-series was interrupted and only previously-collected profiles were retained (i.e. 9837 stations in BOPAD-prof).

**2.4 Bio-optical products within the first optical depth**

The first optical depth ($Z_{pd}$) was calculated as $Z_{eu}/4.6$ (Morel, 1988), where the euphotic depth, $Z_{eu}$, is the depth at which PAR is reduced to 1 % of its value just below the sea surface and was derived from quality-controlled vertical profiles. The procedure by Organelli et al. (2016a) reduced the number of PAR profiles that can be exploited for deriving optical quantities within the first optical depth by about 40 % (e.g. because of atmospheric clouds). Hence, 5748 stations with quality-controlled $Z_{eu}$ and $Z_{pd}$ values were retained and used to compile BOPAD-surf (Fig. 1).

To compute vertical diffuse attenuation coefficients for downward irradiance ($K_d(\lambda)$) and PAR ($K_d(PAR)$) within $Z_{pd}$, each radiometric profile was binned in 1m intervals. $K_d(\lambda)$ and $K_d(PAR)$ values were then derived from a linear fit, and after removal of outliers, between the natural logarithm of the radiometric quantity and depth (in units of pressure) following Mueller et al. (2003). $K_d(\lambda)$ and $K_d(PAR)$ values obtained from linear fits based on less than 3 points or with a determination coefficient ($r^2$) lower than 0.90 were discarded (Organelli et al., 2017).

Values of Chl, FDOM and $b_{bp}(700)$ were also derived, within the first optical depth, from quality controlled vertical profiles. Before computation, any remaining spikes were purged from FDOM quality-controlled profiles by applying first a median filter (5 point window) and then an average filter (7 point window). Similarly, a median filter (5 point window) was applied





to quality-controlled $b_{bp}(700)$ profiles. Finally, Chl, FDOM and $b_{bp}(700)$ profiles were binned in 1m intervals and the average within $Z_{pd}$ was computed.

## 2.5 Satellite data

To demonstrate the applicability of these global BGC-Argo databases, satellite-derived diffuse attenuation coefficients of downward irradiance at 490 nm ($K_d(490)_{sat}$) obtained by the GlobColour project (ACRI-ST, 2015) were downloaded from the web portal http://seasiderendezvous.fr/matchup.php and compared to the in situ BGC-Argo counterparts. $K_d(490)_{sat}$ were obtained, for the period October 2012 to January 2016, from daily Level 3 chlorophyll merged products and using the empirical algorithm by Morel et al. (2007a). Chlorophyll products were merged using MODIS-Aqua and VIIRS Level 3 products (NASA reprocessing R2014.0), see fully detailed merging procedures in ACRI-ST (2015). As statistics of the match-up analysis, the root mean square error (RMSE, units of $m^{-1}$) and the median percentage difference (MPD) were calculated according to Organelli et al. (2016c).

## 3 Quality-controlled vertical profiles

In this section, specific examples of quality-control are presented for each examined variable to provide context for the database. In the case of Chl profiles, three examples extracted from floats operating in different trophic and optical environments are presented (North Atlantic subpolar gyre, Black Sea and South Atlantic subtropical gyre; Fig. 3). The raw North Atlantic profile (Fig. 3a) exhibits non-photochemical quenching (NPQ) at the surface and positive spikes at depth. After the quality control, NPQ is corrected and the positive spikes that are likely related to biological information are retained (Fig. 3a). The Black Sea vertical chlorophyll profile (Fig. 3b) is characterized by a monotonic Chl increase to depth, where the concentration is expected to be null. As Proctor and Roesler (2010) and Xing et al. (2017) stated, the observed Chl increase at depth is due to very high CDOM (which is a consequence of the anoxic conditions prevailing at depth in the Black Sea) and non-algal matter concentrations that can affect the chlorophyll fluorescence signal. After correcting the profile according to Xing et al. (2017), Chl concentrations below 100 m are zero. The profile from the South Atlantic subtropical gyre exhibited only a non-zero dark offset, which was removed in the quality control (Fig. 3c). We recall here that all quality-controlled Chl values are divided by 2 as recommended by Roesler et al. (2017).

Raw FDOM vertical profiles are generally noisy and spiky, especially in the upper water column (Fig. 4). After the quality control, large spikes are identified and removed, and the profile is aligned to match the 950-1000 m median value of the first profile acquired by the float (Fig. 4). No standard correction was applied for water temperature dependence of FDOM (Wratas et al., 2011; Downing et al., 2012), so that it can be applied at the user's discretion. In addition, depending on the application, further processing of FDOM profiles such as smoothing and filtering is recommended before use (see for example Sect. 2.4).





In the case of $b_{bp}(700)$ vertical profiles, the examples in Fig. 5 represent two different steps of the quality-control. The main difference consists in removing positive spikes from the quality-controlled profiles (Fig. 5). Although these spikes likely indicate the occurrence of large aggregates and are essential to monitor carbon fluxes towards the deep ocean (Briggs et al., 2011), they can introduce some noise when export of particulate organic carbon due to small particles (Dall'Olmo and Mork,

2014) or the physiological status of the algal community (Barbieux et al., submitted) are analyzed.

All the quality-controlled profiles of $E_d(\lambda)$ and PAR included in the presented database correspond to Type 1 (i.e. best quality) in Organelli et al. (2016a). The examples in Fig. 6 represent $E_d(412)$ profiles collected in Eastern Mediterranean Sea waters under different sky conditions. The profile in Fig. 6a is acquired under uniform sky conditions. In this case, the quality-control procedure only identifies and removes dark values at depth (not shown) and those corresponding to wave

focusing (Zaneveld et al., 2001) at the surface. The profile in Fig. 6b is instead characterized by non-zero dark values in deep waters (not shown) and sporadic atmospheric clouds. The ensemble of tests of the applied quality-control procedure (Organelli et al., 2016a) detects the various perturbations (Fig. 6b). Correction for dark offset on the sensor's temperature dependence (Mueller et al., 2003) are not performed, so that they can be implemented at the user's discretion. Additional examples showing performances of applied quality-control procedure for $E_d(380)$, $E_d(412)$, $E_d(490)$ and PAR can be found in

Organelli et al. (2016a).

#### 4 BOPAD-prof: spatio-temporal distribution of the biogeochemical and optical Argo database of vertical profiles

Following the main scientific objectives of several international projects (see acknowledgements), the BGC-Argo floats used in these two databases have been primarily deployed in under-sampled areas of key interest for biogeochemical processes such as those with distinct phytoplankton blooms and those with significant export of organic carbon to the deep ocean

(Sarmiento et al., 1992; Takahashi et al., 2002; D'Ortenzio and Ribera d'Alcalà, 2009; Alkire et al., 2012; Lacour et al., 2015). As a result, the 9837 BGC-Argo stations of vertical profiles within BOPAD-prof cover a wide range of trophic conditions, prevailing in open-ocean environments, and represent the first step to set up a publicly available global and inter-operable database for biogeochemical and bio-optical studies. Hereafter, we present the spatial and temporal coverage of quality-controlled vertical profiles for each biogeochemical and bio-optical variable between the world's hemispheres and

among regions. The spatio-temporal distribution of temperature profiles, which are representative of the entire raw database, is also shown.

The latitudinal and monthly distributions of the quality-controlled profiles show similar patterns among the 8 variables (Fig. 7), which indicates that the quality control procedures do not bias the sampling spatially or temporally. However, because of the quality-control procedures, the total number of profiles for a given latitude and month of the year is different among

variables. The number of quality-controlled profiles is generally the highest for chlorophyll (Fig. 7c) and $b_{bp}(700)$ (Fig. 7g). Because of the strict quality-control by Organelli et al. (2016a) that removes radiometric profiles acquired under very unstable meteorological conditions, the total number of $E_d(\lambda)$ and PAR profiles is generally the lowest (Fig. 7i, k, m, o).



In the Northern hemisphere, the database covers a broader latitudinal range than in the South hemisphere. Data range from the Equator to the Arctic Ocean, and late spring to mid-summer are the most represented periods. The number of profiles substantially is lowest between January and April, and especially for radiometric quantities (Fig. 7) as a consequence of the decreasing stability of the water column associated with deteriorated sky and sea conditions (D'Ortenzio et al., 2005; Lacour

et al., 2015). This high contribution of the northern hemisphere to the database is due to the first projects piloting the deployment of BGC-Argo floats that were mainly focused on the North Atlantic subpolar gyre (i.e., 48-65º N; remOcean project) and the Mediterranean Sea (i.e., 31-44º N; NAOS project). Latitudes higher than 67º N are included thanks to a 3-year operating float collecting all variables except FDOM (Fig. 7g). Latitudes between 0 and 30º N (i.e., subtropical gyres and surrounding zones) are also represented owing to measurements acquired by 10 BGC-Argo floats (Fig. 7). Note,

however, that the number of FDOM profiles at these latitudes is lower than for the other variables as a consequence of sensor failure on some floats and absence in those floats deployed in the framework of UK Bio-Argo and E-AIMS projects (where the FDOM sensor was replaced by a sensor measuring particle backscattering coefficient at 532 nm, $b_{bp}(532)$). The northern hemisphere is also represented by data collected in two marginal seas (Fig. 1): the Black and Red seas. Similar to subtropical gyres and surrounding areas, the number of FDOM profiles in the Black Sea is lower than for other variables because half of

the floats deployed in this area measured $b_{bp}(532)$ instead of FDOM.

The southern hemisphere is primarily represented by data collected at latitudes between 38 and 56º S (Fig. 7) in the Atlantic and Indian sectors (Fig. 1). In contrast to the northern hemisphere, no floats have been deployed or reached latitudes higher than 60º S (Fig. 7). Measurements of each variable are also acquired by 7 floats in southern subtropical gyres (around 16-25º S) both in the Atlantic and Pacific Oceans and by 2 floats in the region close to New Caledonia in the South Pacific (Fig. 1).

The temporal coverage of data collected in the southern hemisphere remains uniform from January to September for each variable, but then increases from October to December (Fig. 7). This reflects a switch to adaptive sampling to better resolve the phytoplankton bloom in the southern hemisphere. Similar to the northern hemisphere, the number of radiometric profiles tends however to slightly decrease during the autumn and the Austral summer (from June to August) as a consequence of the worsening meteorological conditions and deepening mixed layer depths (Dong et al., 2008).

The 25 selected regions (and grouped into 9 major areas) contribute, in terms of number of profiles, in different proportions to the database (Fig. 8). This is a consequence of the different number of floats deployed in each area together with a modulated profiling frequency (from 1 to every 10 days). The North Atlantic Ocean dominates BOPAD-prof, as a consequence of the intensive sampling characterizing the subpolar gyre area in multiple programs. Vertical profiles acquired in the Southern Ocean and the Western Mediterranean Sea each represent 18 % on average of the database. The Eastern

Mediterranean Sea is about 14 %, while the South Atlantic subtropical gyre and surrounding areas contribute 6.3 % on average. The South Pacific Ocean represents only 3 to 5 % of the vertical profiles within BOPAD-prof, while polar and marginal seas represent individually a proportion <3 % of each collected variable.



### 5 BOPAD-surf: properties of the bio-optical database within the first optical depth and joint use with remote sensing of ocean color

Because of the unique in situ spatial and temporal coverage, the international community of optical oceanographers (Claustre et al., 2010b; IOCCG, 2011, 2015; Biogeochemical-Argo Planning Group, 2016) has recently recognized measurements
collected by BGC-Argo floats as a fruitful resource of data for bio-optical applications, such as the identification of regions with optical properties departing from the mean statistical relationships (Organelli et al., 2017) as well as the validation of ocean color reflectance (Gerbi et al., 2016) and bio-optical products (IOCCG, 2015). In this context, BOPAD-surf has been compiled with 5748 stations of biogeochemical (i.e., Chl and FDOM) and bio-optical (i.e., $K_d(\lambda)$, $K_d(PAR)$ and $b_{bp}(700)$) variables within the first optical depth (i.e., the layer of interest for ocean color) as derived from previously quality-
controlled vertical profiles. The characteristics of this database are, hereafter, described.

All the 5748 BGC-Argo stations correspond to quality-controlled measurements of euphotic and first optical depths, and represent about 60% of the database of quality-controlled vertical profiles. Global ranges and averages (and associated standard deviations) of $Z_{eu}$ and $Z_{pd}$ and of the other variables are reported in Table 1. In agreement with previous observations (Morel and Maritorena, 2001; Lee et al., 2007; Morel et al., 2007a; Morel et al., 2007b; Soppa et al., 2013;
Organelli et al., 2014), values of $Z_{eu}$ and $Z_{pd}$ vary mostly in the ranges 10.5-180.2 m and 2.3-39.2 m, respectively, with deepest values characterizing the Atlantic and South Pacific Oceans gyres (Fig. 9a, b). Shallowest $Z_{eu}$ and $Z_{pd}$ layers are instead characteristic of the North Atlantic subpolar gyre in spring, the Western Mediterranean and the Black Seas (Fig. 9a, b). The observed ranges of Chl, FDOM, $b_{bp}(700)$, $K_d(\lambda)$ and $K_d(PAR)$ values derived from BGC-Argo measurements (Table 1) are also in good agreement with previous observations (Morel and Maritorena, 2001; Morel et al., 2007a; Morel et al.,
2007b; Cetinić et al., 2012; Dall'Olmo et al., 2012; Peloquin et al., 2013; Sauzède et al., 2015; Valente et al., 2016). As examples of their spatial distribution across the explored regions, $K_d(412)$ and $K_d(PAR)$ are shown in Figures 9c and d, respectively. The reader is referred to the work by Organelli et al. (2017) for regional variability of $K_d(380)$ and $K_d(490)$ coefficients.

As a consequence of the variable-specific quality-control procedures, each variable within BOPAD-surf is represented with
different proportions in the 25 regions (Table 2). Of the 5748 stations with quality-controlled $Z_{eu}$, 83-90 % contain Chl, FDOM and $b_{bp}(700)$ measurements, 62-72 % contain $K_d(\lambda)$ values within $Z_{pd}$, and > 90 % contain $K_d(PAR)$. The Labrador Sea region contains the highest fraction of profiles of each variable (13.81-17.08 %), while the Iceland basin and the Irminger Sea contribute on average 7.6-7.8 % of the profiles in the database (Table 2). In the Mediterranean Sea, the Ionian Sea, Northern and Southern western basins each contribute between 5.5 and 9.7 % of the profiles, while Levantine and
Tyrrhenian Seas each contribute about 4 % on average (Table 2). In the southern hemisphere, the eastern Atlantic and the Indian sectors of the Southern Ocean each contribute about 6-10 % of the entire database, while the relative contribution of the western part of the Atlantic sector is < 4.45 % (Table 2). Subtropical gyres of both hemispheres contribute from 1.47 to 4.43 % according to the variable (Table 2). Marginal seas (i.e. Black and Red seas) and transition zones among various trophic regimes represent less than 3 % of the whole database within the first optical depth (Table 2).





The goal of BOPAD-surf supporting in situ and remote bio-optical applications is demonstrated by two examples of possible use. As a first exercise, previously established bio-optical relationships (Morel et al., 2007a) are evaluated against the BGC-Argo database. It is important to identify the regions with bio-optical behaviors deviating from the average trend, because a bio-optical anomaly could likely lead to uncertainties in retrieving bio-optical and biogeochemical quantities from satellite

ocean color observations (Organelli et al., 2017). The relationship of $K_d(PAR)$ as a function of $K_d(490)$ for the BGC-Argo database is in good agreement with those by Morel et al. (2007a). Slight deviations appear, however, at lowest $K_d(PAR)$ and $K_d(490)$ values and mainly correspond to samples collected in the subtropical gyres and Eastern Mediterranean Sea (Fig. 10a; Table 3). On the contrary, when analyzing $Z_{eu}$ varying as a function of Chl concentrations (Fig. 10b; Table 3), differences appear between the BGC-Argo database and the mean average relationship previously established by Morel et al.

(2007a) using near-surface chlorophyll values. This suggests limited global representativeness of the previous established model, especially with respect to some regions (Organelli et al., 2017). However, it is important to note that deviations may depend also on the uncertainty related to chlorophyll concentrations within BOPAD-surf. Although the spatio-temporal coverage of the Morel et al. (2007a) database is smaller than the BGC-Argo database (Organelli et al., 2017), Morel et al. (2007a)'s previous relationships were based on chlorophyll concentrations determined by High Performance Liquid

Chromatography (HPLC) which is the most accurate technique to estimate phytoplankton pigments. Instead, chlorophyll concentrations within BOPAD-surf are derived from fluorescence measurements calibrated with HPLC. Estimated values, used as a proxy of phytoplankton biomass, can therefore be influenced by regional variations in the fluorescence to chlorophyll ratios, that in turn depend on changes in nutrient availability, growth, photophysiology and taxonomic composition of algal communities (Cullen, 1982), and may not be comprehensively taken into account by the corrections

here applied (Roesler et al., 2017).

In a second exercise, $K_d(490)$ values obtained from the merged GlobColour satellite products (ACRI-ST, 2015) are compared to $K_d(490)$ coefficients obtained from BGC-Argo floats (Fig. 11). While the two products agree approximately at moderate values ($K_d(490) \approx 0.1$ m$^{-1}$) estimates from BGC-Argo floats are considerably lower on average, especially at high and very low water clarity. This result strongly warrants further investigation. Thanks to the unprecedented spatial and

temporal distribution provided by these autonomous platforms, ocean-color algorithm and product validation can routinely be performed in several regions so that errors and possible causes of failure (e.g., influence of Raman scattering; Westberry et al., 2013) can be assessed and/or solved, and algorithms be refined for improving the quality of retrievals.

## 6 Conclusions and recommendations for use

The first measurements of biogeochemical and bio-optical variables collected by the PROVOR-CTS4 generation of

autonomous BGC-Argo floats have been quality-controlled and synthesized in a global database of vertical profiles (BOPAD-prof). Profile-derived bio-optical variables within the first optical depth have been also condensed in a database





dedicated to support field and remote bio-optical applications (BOPAD-surf). Spatial and temporal coverages have been presented and discussed.

The two databases presented here can be directly exploited for several applications, from biogeochemistry and primary production estimation and modeling, to ocean color algorithm and product validation. Online platforms (i.e.

http://seasiderendezvous.eu) are already available to support near real-time ocean-color applications and interactive management of Biogeochemical Argo profiles. In addition, we remind that, according to the specific use intended for these data, further processing may be needed. Additional corrections, e.g. dark counts and temperature dependence for radiometric or FDOM measurements might be required at the user's discretion. Additional or regional adjustments on the calibration factor for chlorophyll fluorescence might be also needed (Roesler et al., 2017). The quality-control procedures applied here

remove only major, known sensor issues.

Finally, these two databases are a first step to provide users with the unprecedented quantity of autonomous in situ measurements processed with common internationally-accepted procedures. However, due to the characteristics of the Biogeochemical Argo network (Johnson and Claustre, 2016; Biogeochemical Argo Group, 2016) and its youthfulness, both databases are likely to evolve as new regions are explored, improved vertical and temporal frequency is achieved, and more

advanced quality-control procedures are developed. Therefore, it is expected that BOPAD-prof and BOPAD-surf could be amended/enriched in the future with new quality-controlled profiles and products, and merged with other already operating configurations of autonomous profiling floats and sensors. The way the two databases have been built makes them potentially fully inter-operable with future databases.

### 7 Data availability

BOPAD-prof (Barbieux et al., 2017) and BOPAD-surf (Organelli et al., 2016b) are publicly available from SEANOE (SEA scieNtific Open data Edition) publisher. Float name, number of cycles and profile, date, latitude and longitude are reported in both databases. In BOPAD-prof, vertical profiles of Chl before quality-control, and $b_{bp}(700)$ with additional removal of spikes (see Sect. 2.4) are also included. BOPAD-surf includes standard errors of $K_d(\lambda)$ and $K_d(PAR)$ as derived from a linear fit (see Sect. 2.5), and standard deviations of averaged Chl, FDOM and $b_{bp}(700)$ values within the first optical depth.   BGC-

Argo raw data used in this study are publicly available online (at ftp://ftp.ifremer.fr/ifremer/argo/dac/coriolis) and distributed as netCDF files.

**Acknowledgments.** This study received funds and support by the following research projects: remOcean (funded by the European Research Council, Grant Agreement No 246777), NAOS (funded by the Agence Nationale de la Recherche in the

frame of the French ''Equipement d'avenir'' program, Grant Agreement No ANR J11R107-F), AtlantOS (funded by the European Union's Horizon 2020 research an innovation program, Grant Agreement No 2014-633211), SOCLIM (funded by the Fondation BNP Paribas), E-AIMS (funded by the European Commission's FP7 project, Grant Agreement No 312642),





U.K. Bio-Argo (funded by the Natural Environment Research Council, Grant Agreement No NE/L012855/1), REOPTIMIZE (funded by the European Union's Horizon 2020 research and innovation program, Marie Sklodowska-Curie Grant Agreement No 706781), Argo-Italy (funded by the Italian Ministry of Education, University and Research (MIUR)), and the French Bio-Argo program (Bio-Argo France; funded by CNES-TOSCA, LEFE Cyber, and GMMC). We thank the PIs of

several BGC-Argo floats missions and projects: Sorin Balan (GeoEcoMar, Romania); Pascal Conan (Observatoire Océanologique de Banyuls sur mer, France; Bio-Argo France); Laurent Coppola (Laboratoire d'Océanographie de Villefranche, France; Bio-Argo France); Kjell-Arne Mork (Institute of Marine Research, Norway; E-AIMS); Anne Petrenko (Mediterranean Institute of Oceanography, France; Bio-Argo France); Pierre-Marie Poulain (National Institute of Oceanography and Experimental Geophysics, Italy; Argo-Italy); Jean-Baptiste Sallée (Laboratoire d'Océanographie et du

Climat, France; Bio-Argo France); Violeta Slabakova (Bulgarian Academy of Sciences, Bulgaria; E-AIMS); Sabrina Speich (Laboratoire de Météorologie Dynamique, France; Bio-Argo France); Emil Stanev (University of Oldenburg, Germany; E-AIMS); and Virginie Thierry (Ifremer, France; Bio-Argo France).

**Competing interest**

The authors declare that they have no conflict of interest.

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

Table 1: Number of profiles, minimum, maximum, average (± standard deviation) values for each variable included in the global Biogeochemical Argo database within the first optical depth (BOPAD-surf).

| Parameter | N | Minimum | Maximum | Average ± standard deviation |
|---|---|---|---|---|
| $Z_{eu}$ (m) | 5748 | 10.5 | 203.8 | 74±32 |
| $Z_{pd}$ (m) | 5748 | 2.3 | 44.3 | 16±7 |
| Chl (mg m$^{-3}$) | 4858 | 0.014 | 12.15 | 0.71±1.21 |
| FDOM (ppb) | 4787 | 0.028 | 4.80 | 1.32±0.76 |
| $b_{bp}$(700) (m$^{-1}$) | 5177 | 0.00009 | 0.0274 | 0.0013±0.0012 |
| $K_d$(380) (m$^{-1}$) | 4156 | 0.015 | 0.520 | 0.103±0.065 |
| $K_d$(412) (m$^{-1}$) | 3951 | 0.010 | 0.546 | 0.090±0.059 |
| $K_d$(490) (m$^{-1}$) | 3553 | 0.017 | 0.475 | 0.065±0.046 |
| $K_d$(PAR) (m$^{-1}$) | 5245 | 0.025 | 0.464 | 0.092±0.051 |





**Table 2: Relative contribution (%) of biogeochemical and bio-optical variables for the 25 geographic regions included in the Biogeochemical Argo database within the first optical depth (BOPAD-surf).**

| Region | Basin | Chl | FDOM | $b_{bp}(700)$ | $K_d(380)$ | $K_d(412)$ | $K_d(490)$ | $K_d(PAR)$ |
|---|---|---|---|---|---|---|---|---|
| Arctic Sea | Norwegian Sea | 1.54 | 0.00 | 1.60 | 1.52 | 1.95 | 1.49 | 1.58 |
| Black Sea | Black Sea | 2.68 | 0.00 | 2.07 | 2.38 | 2.96 | 2.73 | 2.21 |
| Western Mediterranean Sea | Northwestern | 8.79 | 9.71 | 9.02 | 7.89 | 7.87 | 7.97 | 8.01 |
| | Southwestern | 7.62 | 8.13 | 7.11 | 7.24 | 5.59 | 5.88 | 6.58 |
| | Tyrrhenian Sea | 3.83 | 4.97 | 4.15 | 4.55 | 3.32 | 2.98 | 3.85 |
| Eastern Mediterranean Sea | Ionian Sea | 7.60 | 9.34 | 5.52 | 9.24 | 8.00 | 8.53 | 8.27 |
| | Levantine Sea | 2.96 | 5.39 | 5.62 | 4.21 | 3.32 | 3.80 | 4.67 |
| North Atlantic subpolar gyre | Labrador Sea | 15.62 | 13.81 | 15.14 | 15.18 | 17.08 | 17.00 | 14.41 |
| | Irminger Sea | 7.64 | 8.36 | 7.78 | 7.87 | 8.83 | 7.15 | 7.42 |
| | Iceland Basin | 7.00 | 7.19 | 7.48 | 7.56 | 8.86 | 8.19 | 7.15 |
| | South Labrador Sea | 0.58 | 0.00 | 0.52 | 0.51 | 0.71 | 0.56 | 0.51 |
| | Transition zone | 0.95 | 0.96 | 0.89 | 0.82 | 0.81 | 0.90 | 0.84 |
| North Atlantic subtropical gyre | Subtropical gyre | 2.26 | 2.32 | 2.86 | 2.26 | 1.97 | 2.56 | 2.61 |
| | Eastern subtropical gyre | 1.07 | 0.00 | 0.00 | 1.32 | 1.01 | 1.27 | 1.35 |
| | Western subtropical gyre | 0.08 | 0.00 | 0.10 | 0.05 | 0.00 | 0.03 | 0.08 |
| | Transition zone | 1.01 | 0.00 | 1.00 | 0.12 | 0.28 | 0.45 | 0.82 |
| Red Sea | Red Sea | 0.93 | 1.04 | 1.04 | 0.99 | 0.66 | 0.65 | 0.78 |
| South Atlantic Ocean | Subtropical gyre | 1.63 | 4.43 | 4.10 | 3.73 | 2.86 | 3.26 | 3.74 |
| | South subtropical gyre | 0.33 | 0.00 | 0.95 | 0.77 | 0.58 | 0.84 | 0.90 |
| | Transition Zone | 1.81 | 0.00 | 1.70 | 0.99 | 1.47 | 1.60 | 1.51 |
| Southern Ocean | Atlantic sector | 3.54 | 3.91 | 0.64 | 4.45 | 2.91 | 3.74 | 3.68 |
| | Atlantic to Indian sector | 10.09 | 10.28 | 9.54 | 7.84 | 10.07 | 8.13 | 8.77 |
| | Indian sector | 7.93 | 6.25 | 7.51 | 6.09 | 6.73 | 7.60 | 7.11 |
| South Pacific Ocean | Subtropical gyre | 1.56 | 2.74 | 2.53 | 1.56 | 1.47 | 1.77 | 2.21 |
| | New Caledonia | 0.97 | 1.19 | 1.12 | 0.87 | 0.68 | 0.90 | 0.93 |

5   **Table 3: Statistics of regression fits displayed in Figure 10. Standard error of each coefficient is shown in parentheses.**

| Equation | N | a | b | c | d | $r^2$ |
|---|---|---|---|---|---|---|
| $K_d(PAR)= a+b*K_d(490)+c*K_d(490)^{-1}$ | 3401 | 0.062 (0.002) | 0.869 (0.011) | -0.001 ($4*10^{-5}$) | - | - |
| $\log_{10}(Z_{eu})= a+b*\log_{10}(Chl)+c*\log_{10}(Chl)^2+d*\log_{10}(Chl)^3$ | 4858 | 1.688 (0.003) | -0.348 (0.006) | -0.140 (0.009) | -0.017 (0.005) | 0.57 |



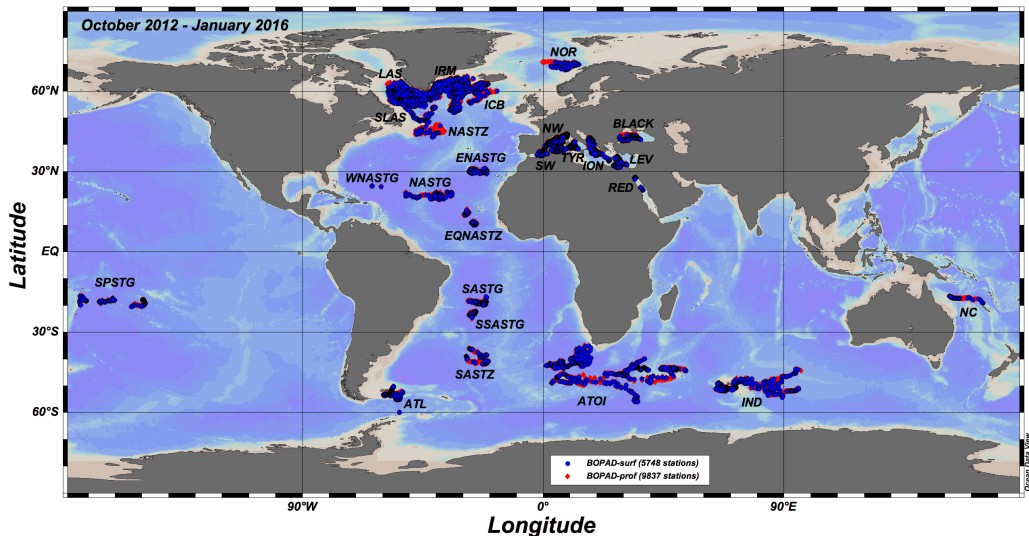

**Figure 1: Location of the 9837 stations collected by 105 Biogeochemical Argo floats in the period October 2012 – January 2016 that compose the database of vertical profiles (BOPAD-prof). Dots indicate the 5748 stations used to assemble the database within the first optical depth and devoted to bio-optical applications (BOPAD-surf). Abbreviations for the 25 geographic regions used to group the stations are also displayed (see Table S1 in supplement material for full description). The map is drawn by the Ocean Data View software (R. Schlitzer, Ocean Data View, http://odv.awi.de).**

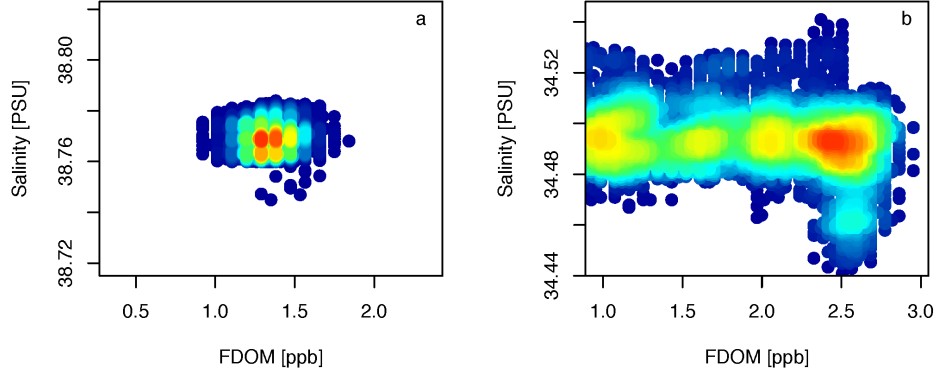

**Figure 2: Salinity (psu) vs FDOM (ppb of quinine sulphate) data collected during drift at 1000 m for two Biogeochemical Argo floats: a) Float WMO 6901768 (Eastern Mediterranean Sea) showing no FDOM changes for a similar salinity (7 months of sampling); b) Float WMO 6901439 (South Atlantic subtropical gyre) showing a decrease of FDOM for a similar salinity (more than 12 months of sampling). In plot b, FDOM values around 2.5 ppb of quinine sulphate represent measurements collected during the first months of the float lifetime. Colours indicate density of measurements for a given salinity vs FDOM value (red>blue).**




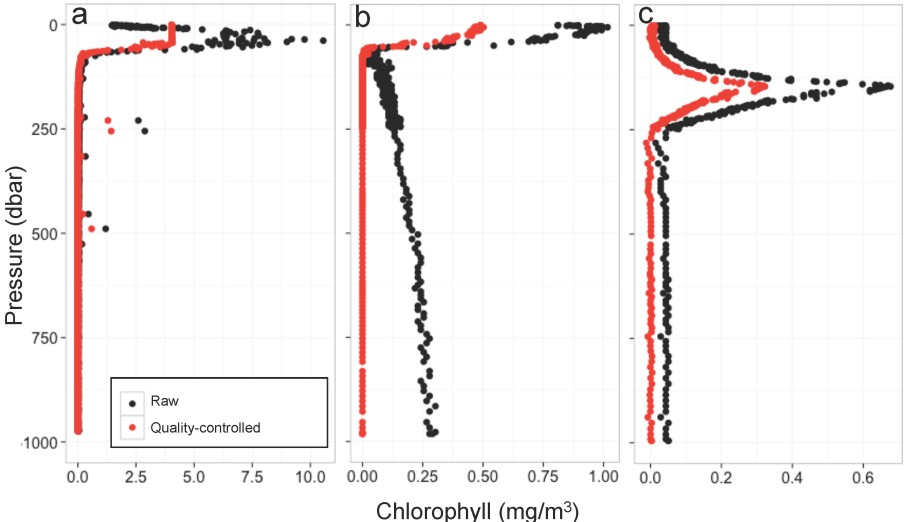

**Figure 3: Raw and quality-controlled vertical profiles of chlorophyll *a* concentration (Chl) for the following areas: a) the North Atlantic subpolar gyre (float WMO 6901516); b) the Black Sea (float WMO 7900591); c) the South Atlantic subtropical gyre (float WMO 6901439).**

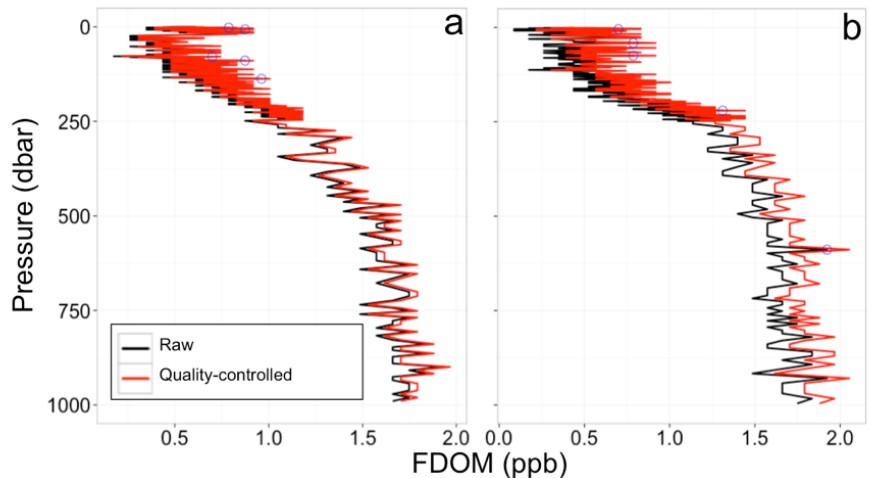

**Figure 4: a, b) Raw and quality-controlled vertical profiles of fluorescent dissolved organic matter (FDOM, ppb of quinine sulphate) collected by the profiling float WMO 6901440 in the South Atlantic subtropical gyre. Blue open circles indicate positive spikes.**



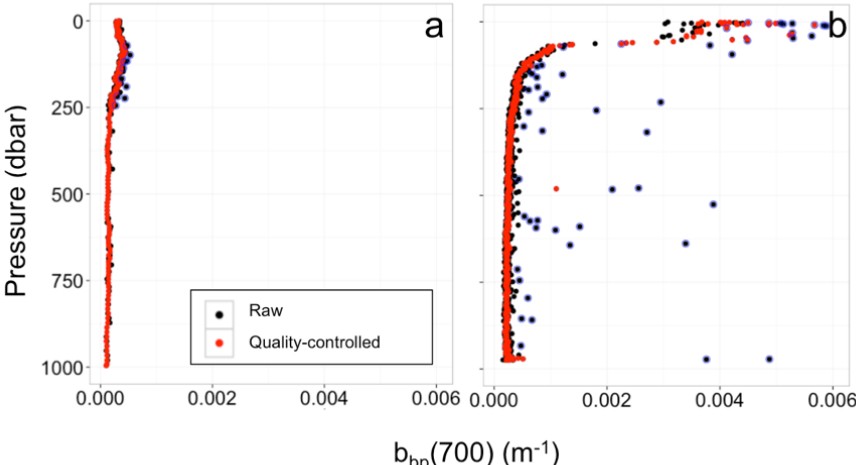

**Figure 5: a, b)** Raw (i.e. before spike removal) and quality-controlled (i.e. after spike removal and application of an average filter) vertical profiles of particle optical backscattering at 700 nm ($b_{bp}$(700)) collected in the South Atlantic subtropical (float WMO 6901439) and North Atlantic subpolar (float WMO 6901516) gyres, respectively. Blue open circles indicate positive spikes.

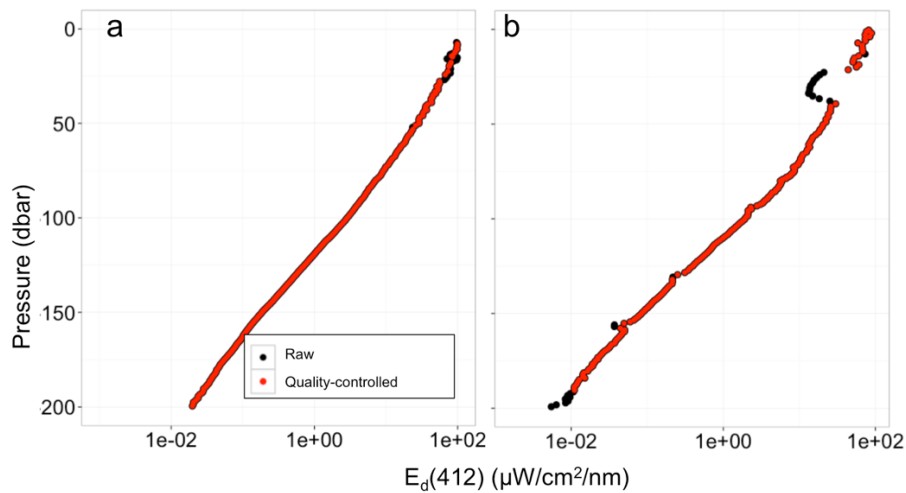

**Figure 6:** Raw and quality-controlled vertical profiles of downward irradiance at 412 nm ($E_d$(412)) collected by the same profiling float (WMO 6901528) in the Eastern Mediterranean Sea under (a) clear and (b) cloudy sky.



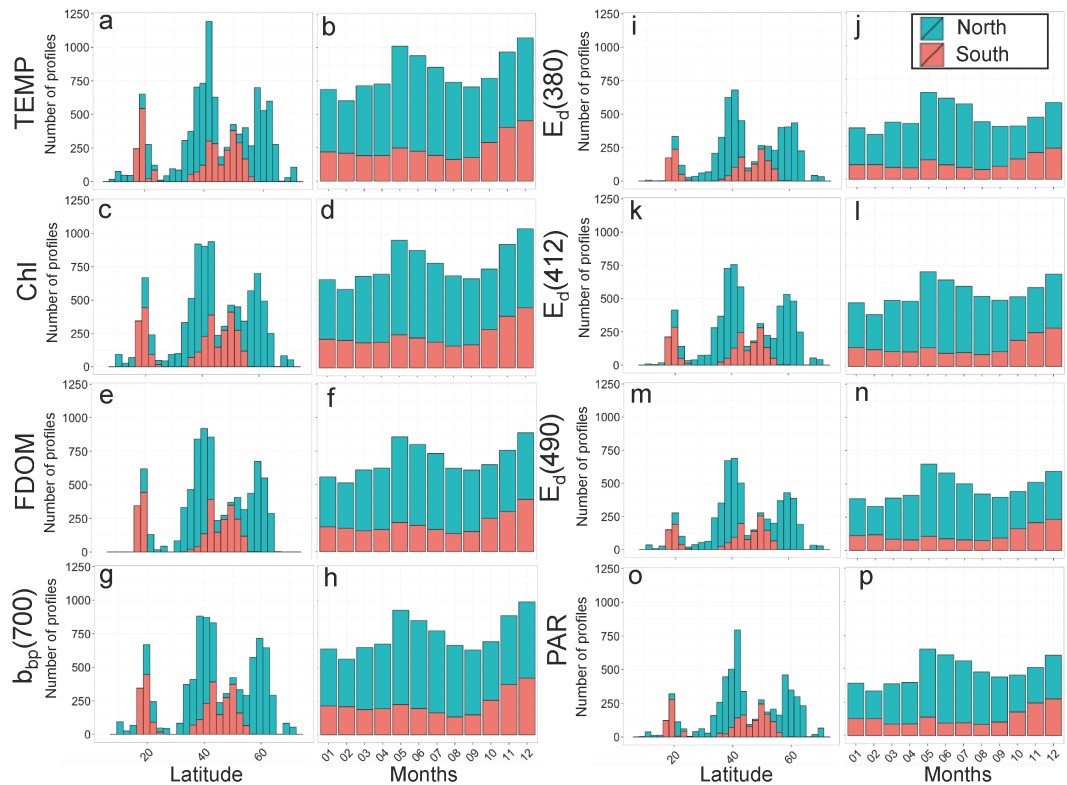

**Figure 7: Latitudinal and monthly distributions of the 9837 vertical profiles, presented as stacked histograms, for: a-b) Temperature (T); c-d) Chlorophyll concentration (Chl); e-f) Fluorescent dissolved organic matter (FDOM); g-h) Particle optical backscattering coefficient at 700 nm ($b_{bp}(700)$); i-j) Downward irradiance at 380 nm ($E_d(380)$); k-l) Downward irradiance at 412 nm ($E_d(412)$); m-n) Downward irradiance at 490 nm ($E_d(490)$); o-p) Photosynthetically available radiation (PAR). Vertical profile distributions are displayed for both northern and southern hemispheres.**





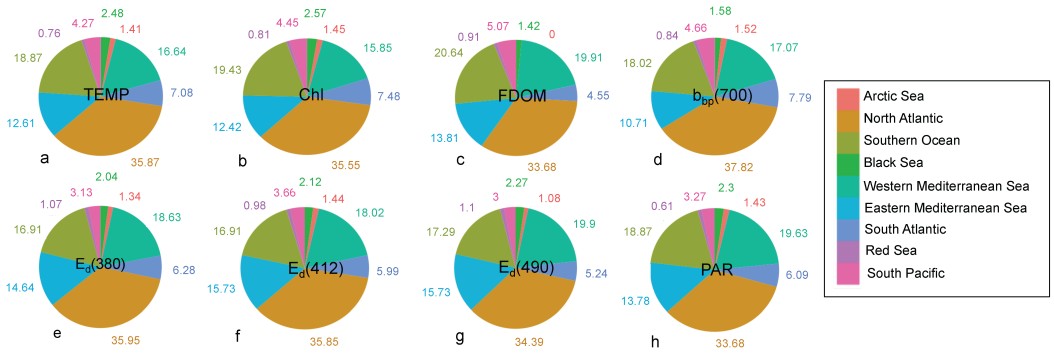

**Figure 8: Relative contributions (%) of the 9837 vertical profiles among 9 regions and sub-regions sampled by Biogeochemical Argo floats: a) Temperature (T); b) Chlorophyll concentration (Chl); c) Fluorescent dissolved organic matter (FDOM); d) Particle backscattering coefficient at 700 nm ($b_{bp}$(700)); e) Downward irradiance at 380 nm ($E_d$(380)); f) Downward irradiance at 412 nm ($E_d$(412)); g) Downward irradiance at 490 nm ($E_d$(490)); h) Photosynthetically available radiation (PAR). See Table S1 (supplement material) for basins included within each region/sub-region.**

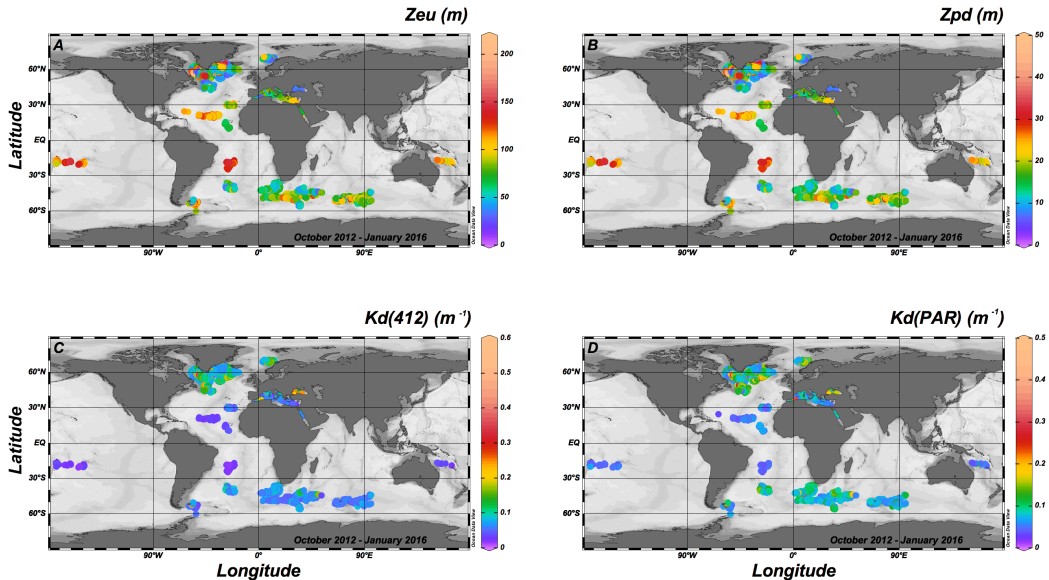

**Figure 9: Global distribution of: a) Euphotic depth ($Z_{eu}$); b) First optical depth ($Z_{pd}$); c) Average value of the diffuse attenuation coefficient of downward irradiance at 412 nm within the first optical depth ($K_d$(412)); d) Average value of the diffuse attenuation coefficient of the photosynthetically available radiation within the first optical depth ($K_d$(PAR)).**





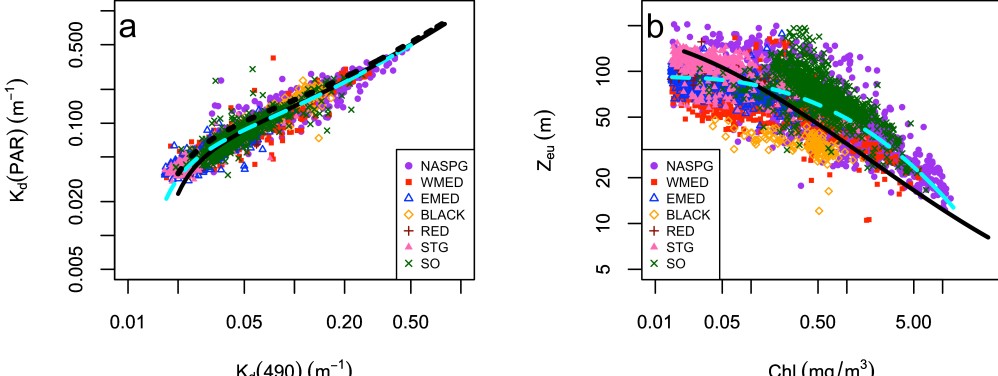

**Figure 10: a) Log-log plot of the diffuse attenuation coefficient for PAR ($K_d$(PAR)) as a function of the diffuse attenuation coefficient for downward irradiance at 490 nm averaged within the first optical depth ($K_d$(490)). The dashed line is the fit to all data. Dotted and solid lines represent relationships established by Morel et al. (2007a; Eq. 9 and 9', respectively) limited to the**
5 **range of $K_d$(490) found in that study; b) Log-log plot of the Euphotic depth ($Z_{eu}$) as a function of chlorophyll *a* concentration (Chl) within the first optical depth as derived from Biogeochemical Argo float measurements. The dashed line is the 3-order polynomial fit to all data. The solid line represents the regression model established by Morel et al. (2007a; Eq. 10) limited to the range of near-surface Chl concentrations found in that study. In both panels, Biogeochemical Argo data are grouped in 7 major areas: Norwegian Sea, North Atlantic subpolar gyre and surrounding areas (NASPG); Western Mediterranean Sea (WMED); Eastern**
10 **Mediterranean Sea (EMED); Black Sea (BLACK); Red Sea (RED); Subtropical gyres and surrounding areas (STG); Southern Ocean (SO). Statistics for Biogeochemical Argo derived fits are in Table 3.**

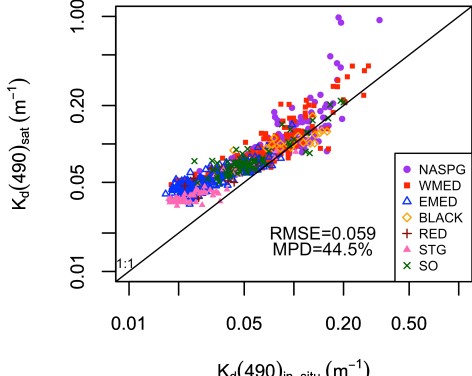

**Figure 11: Comparison (n=658) between the diffuse attenuation coefficient for downward irradiance at 490 nm as derived from satellite measurements ($K_d$(490)$_{sat}$) as a function of $K_d$(490) derived from Biogeochemical Argo float measurements within the first**
15 **optical depth ($K_d$(490)$_{in\_situ}$). The solid line represents the 1:1 line. Biogeochemical Argo data are grouped in 7 major areas: Norwegian Sea, North Atlantic subpolar gyre and surrounding areas (NASPG); Western Mediterranean Sea (WMED); Eastern Mediterranean Sea (EMED); Black Sea (BLACK); Red Sea (RED); Subtropical gyres and surrounding areas (STG); Southern Ocean (SO). The root mean square error (RMSE, units of m$^{-1}$) and the median percentage difference (MPD) for all data are shown.**