# Peer review of "Two databases derived from BGC-Argo float measurements for biogeochemical and bio-optical applications at the global scale"

_Earth System Science Data, 2017_

## Referee Comment (RC1) · Anonymous Referee #1 · 8 Aug 2017

Review ESSD-2017-58, BGC Argo

The authors / data providers have prepared and presented a potentially useful data set. A user gets easy access and very good descriptions. Researchers have good options to use the data in depth profile formats or to take advantage of the euphotic zone summaries. The authors have performed a valuable service by compiling these particular bio-optical data from the larger ARGO data set and by specifying careful but consistent quality control procedures. The data seem like a very good fit to this journal. I applaud the effort and recommend publication. (Having Roesler et al. 2017 in open access proved very important for evaluation of this data set.)

I recommend changes to the presentation of this data that should make it much more useful. I make three over-arching suggestions followed by a sequence of line-specific technical suggestions or questions.

1) Global impact

The data set derives from 105 profilers operating over a time period of 40 months. Although one would have to dig deeply through the data to confirm, I believe that no single profiler operated for the entire 40 months. We have instead a compilation based on a series of relatively short (20 to 30 month) deployments each in a relatively restricted region. Although the map presented here as Figure 1 and the number of "stations" approaching 10'000 seem impressive, on the scale of an evolving global ocean they give us only a snapshot. A useful, unique, challenging snapshot, hard-won in the face of funding and operational constraints, but a relatively short Atlantic-focussed glimpse none-the-less. This data set basically misses most of the Indian and almost the entire Pacific Ocean and, not surprisingly, stays well away from ice-influenced regions. We should feel very well served to have these data! But we should not pretend that they provide us an encompassing view of an evolving global ocean. These authors hint at these limitations in their conclusions (page 11, line 15), where they mention " new regions", "improved vertical and temporal frequency", etc. Against these cautions, I feel that the title which includes the phrase "biogeochemical and bio-optical applications at the global scale" greatly overstates the impact.

The authors might consider several other efforts to compile global biologically-relevant or carbon cycle-relevant data sets for the oceans, including Peloquin et al. (HPLC chlorophyll) and Valente et al. (near-surface bio-optical properties) - both of which they do cite - or Sauzéde et al. (in situ fluorescence) or Bakker et al. (SOCAT, surface ocean CO2) which they do not cite. Taking only those four examples (all from ESSD), we typically see data collected over 2 to 6 decades presenting several 10s of thousands (up to millions, for SOCAT) of stations, profiles and measurements. In this paper we see nearly 10'000 observation in only 40 months (closer to 5'000 for the first optical depth) - the promising impact of ARGO technology which the authors could highlight more clearly - but also clear temporal and spatial limitations compared to other data compilations. In presenting their very careful quality control discussion, these authors have failed to show us clearly what their bio-ARGO profilers have achieved for ocean observations and also how these data contribute to, fit with, supplement, or surpass prior and on-going pan-oceanic data compilations. For this reader, these authors have missed an opportunity to quantify how "The BGC-Argo sampling approach can therefore help the scientific community accumulate observations on biological and biogeochemical properties of the ocean" (page 2, line 10). We want to do more than merely "accumulate"?

2) Representativeness

From the understandable view of these biological and bio-optical oceanographers, the ideal ocean situation occurs when a profiler reaches the ocean surface near local noon with small waves and a cloud-free sky. In this paper we encounter a series of qualitative statements about variances from those ideal conditions:

Page 7, line 33 "unstable" meteorological conditions.
Page 8, line 4 - Again a focus on stability (now of the water column!) and "deteriorated sky and sea conditions".
Page 8, line 24 "worsening meteorological conditions and deepening mixed layer depths".

But, the global ocean represents a windy, cloudy, stormy place.  Large regions have persistent coverage of stratus clouds at certain times of the year.  For some parts of the ocean we have almost no cloud-free images despite nearly 40 years of daily satellite observations.  From a biological view, we should appreciate disturbed conditions and vigorous mixing processes.  I believe the standard (non-biological) ARGO profilers rise to the surface without regard to sky or wind?  If, in this data set, the authors, consciously or sub-consciously, focus on and lead the user toward mid-day profiles under calm seas and clear sky conditions, we together suffer the risk of developing a serious bias?

3)  Accuracy

Having overcome many of the serious technical, operational and funding challenges of gaining useful bio-optic data from autonomous ocean profilers, and having applied a consistent set of quality control procedures intended primarily to remove spikes and outliers (and secondarily to identify instrument drift) the authors then present all data as uniformly certain.  Although the authors show data distributions in several plots, we see no error bars and no uncertainty shading.  For intercomparisons, these authors give us only global ranges (min-max values).

The description needs to thoroughly address uncertainty in a specific section.  After all processing, what remaining uncertainties apply to what data? What changes in operation, instrumentation or data processing could or could not address those uncertainties?  For the derived properties $Z_{EU}$ and $Z_{PD}$ we get statistical (standard deviation) uncertainties (e.g. Table 1 on page 17) but those derive from the data processing and do not address uncertainties in the underlying measurements?  For many real-world measurements uncertainty remains very difficult to quantify but in this case the reader needs from the authors at least a sense of confidence and uncertainty for each product.  If, as the authors clearly hope, these data prove useful for global models, quantitative uncertainty information will prove an absolute requirement.

These authors need to explicitly address issues raised by Roesler et al.  In its present form, this manuscript appears to have added those issues, and applied a uniform 2x correction, after the fact or at least late in the preparation process.  Roesler et al attempted to exclude from their analysis exactly those mid-day, clear sky, high irradiation conditions that this compilation seems to favour - how does that mismatch affect the values presented here?  Roesler et al - using many of these same data! - showed a very strong regional dependence of correction factors, indicating that a global uniform application of a 2x correction factor would in fact prove seriously wrong in almost all cases for almost all regions.  Yet here we read about a simple 2x correction for chlorophyll values?   Roesler et al. provided explicit regional- or biome-based correction factors that this paper should have considered?

In its present form, without an explicit discussion of uncertainty, this manuscript will fail to meet the needs and expectations of many potential users.

Specific comments as follows:

Data considerations

Why, at http://www.seanoe.org/data/00360/47142/ (corresponds to doi http://doi.org/10.17882/47142) do we find two versions, version 1 and version 2?  Readme text in version 2 explains the differences, but the deletion of these 20-some profiles received no mention or justification in the text.  Properly, a doi should point exclusively to a single version of any data.  Second version should carry a second doi.

We get profile data in separate parameter-based files (e.g. CHLA, CDOM), space delimited. We get the derived optical depth data in one file, properly comma delimited. (.csv).  Why do we not get all files in .csv format?

Floats to breach surface around local noon. 100 floats times 100 profiles per each gives roughly 10000 profiles. At once per 10 days, 100 profiles would take 1000 days, not quite 3 years. Average profile interval must therefore approach more like 13-15 days? No profiler operates for the full 40 months, but wrong to specify an average of 10 days over that full time period. More properly an average of 10 days while operating?

Page 2, Lines 16, 17: "nitrate concentrations" Has any ARGO biogeo float solved the NO3 challenge? Not sure why the authors mention this?

Page 4, correcting the FDOM. Removed spikes outside 25 and 75, and then additionally remove spikes greater than 4 x the mean value? Identification and quantification of temporal deterioration assumes consistency of deep water masses?

We need to know the duration of each profiler's operation. Can we determine the average lifetime of each float? E.g float 7900591 operated from 2013-12-20 to 2015-07-05, e.g. 30 months, over which time it took roughly 70 profiles (average interval of 13 days but even greater because for the first month after deployment it apparently profiled once per day). We also need number of profiles per profiler? To understand anything about contamination or other performance deterioration as a function of cumulative time in the water, we need to know average deployment duration for the profilers as well as number of profiles by each. Not hard to extract and perhaps graph this information? This information would also prove helpful in making the points about large effort and great resources needed to achieve even this level of spatial and temporal coverage.

Page 5, line 2: " $E_d$ $(0_-)$" and Page 5, line 9: " $E_d$ $(0_+)$" typos? How related to $E_d$ (lambda )?

Page 5, line 17, tracking possible biofouling. Interruption of the time series occurred in real-time or during post-processing?

Page 6, line 2 - simple vertical average of CHLa, CDOM, etc., for optical depth? How did these optical depths compare to other data, globally or regionally?

Page 6, comparison with satellite data.

The authors have missed an important opportunity to connect and compare these data to the bio-optic data reported by Valente et al. These authors cite that data set (page 9, line 20), but only once for the purpose of confirming the bio-optic properties reported here (and buried in a summary sentence in which the reader can't determine which external paper connected to which parameter reported here). In fact the Valente et al data represent an important partner for these data. Those data stop at 2012, these data extend through 2015. This data includes Labrador Sea and Southern Ocean locations missing in the Valente et al. data. That data has many more values, e.g. for Chl A, that could give a much better regional comparison (North Atlantic, for example) for these data. These authors do not need to do or show the work of attempting to merge this data into a Valente et al. framework but they should explicitly outline the connection points and opportunities. These authors could also very much learn from the graphic approaches (e.g. Valente et al. figure 3 and figure 10) and users of both data sets need some convergence of variable names and units (e.g. for FDOM treated very differently in the two data sets).

Page 7, line 22 "open-ocean environments" If by "open-ocean" the authors mean 'collected in areas with depths greater than 1000 meters, as opposed to shallower continental shelf regions, then we can perhaps accept their definition so long as they describe what they mean more carefully. If, however, "open-ocean" should imply a broad spatial coverage of large ocean regions, then the absence of measurements from the Pacific stands out. The authors can correctly say `within limitations of project-driven resources, deployments focused on some of the important carbon-export regions of the Atlantic Ocean, in all cases in regions with depths greater than 1000 meters'?

Page 8, line 31.  And North Pacific equals 0 %.

Page 9, line 16.  I don't believe, from this data set, the authors can make any statistically-valid statement with respect to any part of the Pacific.

Page 9, final paragraph.  Again, a good description of percentages within regions of deployments, but must include recognition of very large areas (e.g. Pacific) with no deployments.

Page 19, legend to Figure 1.  Dot and diamonds hard to distinguish, should specify red diamonds and blue dots.

Page 19, Figure 2.  Reader to assume that data from float 6901439 were truncated at some point, with data after that time point discarded?

Page 20, Figure 3.  Panel A - Why assume only non-photochemical quenching?  Does NPQ include by definition active surface avoidance?  Panel C includes the 2x factor?

Page 20, Figure 4.  Why the blue circles indicate spikes in corrected (red) curves?  Also, in this case, we assume corrected greater than raw due to a sensor performance issue tracked from the deep FDOM?  But at what point would biofouling or sensor deterioration have disqualified the measurements?

Page 21, Figure 5.  My old eyes see blurred blue dots, no blue open circles.

Page 21, Figure 6.  Ascent speed, profile time?  Variability of cloud shading over that time period?

Page 23, Figure 8 - Not a useful way to show geographic data, confusing.  Use maps instead, as in Figure 9 (or Figure 10 of Valente et al.)

Supplement consists of a single table defining the acronym codes for specific ocean regions. If, as I suspect, it applies to or derives from a larger ocean regional description scheme, the authors should cite the external references.  If it represents a product custom to ARGO generally or bio-ARGO specifically, the authors should include the table as an appendix in this ESSD manuscript. Otherwise, according to Copernicus archive procedures as this reviewer understand them, the archive process could preserve the manuscript but not the supplement.

---

## Referee Comment (RC2) · 23 Aug 2017

The processes and dynamics that define the climate sensitivity of the biological carbon pump are not well understood. This is due in part to our lack of understanding of this complex problem through chronic under sampling of the world's oceans, which do not resolve inter-annual variability and seasonal and intra-seasonal dynamics. Autonomous technology promises to overcome the space-time gap in ocean observations with bio-optical sensors on platforms that are able to profile the water column providing highly cost-effective measurements at high frequency that can characterise the vertical biogeochemistry at smaller scales, but also for sufficiently long periods that may help

to reduce uncertainties associated with carbon budgets at longer time scales. As such, I recommend this highly useful data set for publication and commend the efforts of the authors in all the steps that such an achievement requires; from securing the funds to purchase the numerous floats to arranging for their deployment in a globally diverse manner all the way through to the significant efforts in processing and collating the data into a succinct repository.

However, although I see very obvious benefits in the use of such a database both for ocean colour product validation and to further our understanding of ecosystem dynamics, I have one major concern with regards to utilising the chlorophyll (chla) data for validating ocean colour. The uncertainties in the BGC-Argo chla data are typically large and poorly characterised – often larger than the satellite derived chla estimates (mainly due to the globally applied factor of two bias in the conversion of fluorescence to chl and the simple quenching correction which is difficult to evaluate without night time profiles). This raises some serious concerns with the use of float derived chl a data for match-up based validation application with regard to uncertainty budgets. That being said however, I do not have a problem with the use of the other bio-optical variables (e.g. Kd, bbp, Zeu) for ocean colour validation, which are not susceptible to the same kinds of mismatches in the uncertainty budgets.

Inline with the above, I would recommend some changes to the manuscript that need to be addressed before being suitable for publication and provide some suggestions to improve the database. In addition, I provide a list of minor corrections and suggestions to improve the manuscript and attach a pdf with detailed comments and typos.

1. Major comments

1.1. Using BGC-Argo chla data for ocean colour validation

Although none of the BGC-Argo chla versus satellite chla matchups are presented in the manuscript, the implications to do so for validation purposes are implicit both in the text. (e.g. pg 3 line 4: "data presented in BOPAD-surf are compared with existing biooptical models and used in conjunction with products derived from satellite platforms in order to show applicability for validating ocean-color bio-optical products at the global scale" and pg 9 line 5: "measurements collected by BGC-Argo floats are a fruitful resource of data for bio-optical applications ….. as well as the validation of ocean color reflectance (Gerbi et al., 2016) and bio-optical products (IOCCG, 2015)" and pg 10 line 25 "…ocean-color algorithm and product validation can routinely be performed in several regions so that errors and possible causes of failure …. can be assessed and/or solved, and algorithms be refined for improving the quality of retrievals." ) and even more so in the data base itself (see http://seasiderendezvous.fr/matchup.php) where chla is the default product for match up locations and the colours of the data points represent the % relative error between float and satellite chla matchups. Given the inaccuracies in the float chla data I am not convinced that such a comparison is meaningful, in particular without any indication of the errors implicit in the BGC-Argo chla data. That being said however, I do not feel that the inaccuracies in the chla data render them ineffectual, on the contrary, these data will provide extremely useful information towards an improved understanding of the biological response to physical drivers and our understanding of the sensitivity of the biological carbon cycle to climate change that will ultimately lead to improved estimates of long term trends. For example, although the Southern Ocean bias in satellite estimates of chlorophyll is well known it does not render the data any less useful, it is however important that the user is well aware of the quantitative limitations of the data.

From my understanding, one of the primary drivers of the errors in BGC-Argo chla is the variable relationship between fluorescence and chla which is not accounted for in the quality control step that divides all chla data by a factor of two to correct for the global bias in the factory calibration. Although Roesler et al., 2017 recommends to do so in order to improve the global accuracy of chla measurements from WET Labs ECO sensors, they acknowledge the regional variability in this factor, which ranges from 0.56 in the Arabian Sea to 7.75 in the Southern Ocean. As such, the global application of a factor of two can create errors that range from an underestimate of "actual" chlorophyll

by ±100% in the Arabian Sea to an overestimate of chlorophyll in the Southern Ocean by ±250%. Would it not be possible to use some of the regional variability evident in the relationship between chla from HPLC and ECO-fl (Roesler et al., 2017, their Figure 1) to derive a more regionally robust factor for correcting the factory calibration bias?

**1.2. Quenching correction**

Another area that can introduce a significant amount of error into both the profile and the surface chla data is the choice of quenching correction that is applied. The Xing et al., 2012 method of correcting quenching is robust and effective, so long as the assumptions it relies on are valid. The Xing et al. (2012), method relies on the assumption that a) chlorophyll concentrations within the mixed layer are uniform and b) that quenching processes do not affect depths below the depth of maximum fluorescence within the MLD. This method does not allow for sub surface fluorescence maxima to occur within the mixed layer. The method of Biermann et al. (2015) attempts to overcome this limitation by instead finding the maximum fluorescence within the euphotic layer and extrapolating this value to the surface. A comparison of their method with that of Xing et al. (2012) identified occasions (when the MLD was deeper than the euphotic depth) where quenching was corrected without masking subsurface fluorescence signals. However, as with the method of Xing et al. (2012), this method assumes homogeneity, but in this instance within the euphotic zone as opposed to the mixed layer (i.e. it does not allow for daytime subsurface maxima to be present within the euphotic zone). When these assumptions are not met (i.e. chlorophyll is not homogenous within either the mixed layer or the euphotic layer and quenching occurs below the mixed layer) the result will be a typical underestimate of daytime surface chla in the case of Xing et al. (2012) and an over correction of surface chla when the Biermann et al. (2015) method is applied. As mentioned in Xing et al, 2012, for multi-instrumented platforms with both fluorometers and backscattering sensors, Sackmann et al. (2008) proposed an elegant method that made use of the backscattering profile (as independent proxies of phytoplankton distribution) to correct the fluorometric one.

[Figure]

However, this method still relies on certain assumptions such as a regular association between particulate backscattering and chlorophyll concentration. Regardless, if both backscatter and fluorescence sensors are available then methods that utilise both parameters are perhaps more likely to retrieve accurate estimates of chlorophyll during the day.

Either way, it seems to me that there are at least two other methods of correcting quenching which ought to be applied to your data set and the results compared to try to determine which is the best method to use and when. Or at least have an idea of the different surface chla concentrations that the different methods produce in order to get a handle on the possible range of error that this quality control step can introduce. A major problem with having all the profiles in the data set being performed at midday (apart from the obvious issues with quenching) is that it is very difficult to quantify whether or not a daytime profile has been corrected correctly. As such, I would recommend that future float missions consider doing both midnight and midday profiles in order to improve the quality of the chla data (even if this means a reduction in the longevity of the float life span).

1.3. Error estimates

Given that validation is a quantitative assessment of uncertainty and that the BOPAD-surf data set is intended to be used for satellite validation, I feel that it is important to provide some indication of the anticipated errors in the derived variables. If, in the case of chl, you are wanting to validate a satellite product to within $\sim$35% uncertainty (in Case 1 waters) then it is important to know when your in situ product has an error of >100%. A short quantitative analysis of the expected uncertainties in the float derived chla data would be very useful and is necessary here. Similarly, I think that a more open discussion is required around the limitations and weaknesses of the published database together with its strengths.

2. Minor comments

Page 4 line 5: What about positive spikes in chla? I appreciate that they were retained as they could represent "real" data. However they appear to have been removed from the BOPAD-surf data base. If so please provide details. Also, if the spikes were remove, were they interpolated over in the vertical or left as NaN's?

Page 4 line 10: is it possible to please clarify how you systematically determined the profiles that were affected by non algal fluorescence with depth? e.g. an increase in chla with depth for how many meters beyond what threshold depth? how do you ensure that you are correcting for non-algal increases of fluorescence with depth and not "real" subsurface increases in chla e.g. via a subducting water mass?

Page 5, line 11, step 3: this step is not clear to me? it has already been implied that positive spikes in some data were retained (e.g. chl and bbp) as they can represent "real" information. As such it is not clear to me how you used sharp gradients with depth to test for instrument drift?

Page 5 line 21: how did you get the PAR value just below the surface? please provide method? e.g. fitting an exponential Page 6, line 1: was a similar median filter applied to the chla data to remove positive spikes from the chl data set? If so please describe the method used.

Page 6, line 15, figure 3: please include the MLD and Ed on the example profiles

Page 6. Line 23, figure 3c: t is not clear to me how it is possible to retain this shape of profile if the Xing quenching method is applied. unless the MLD is very shallow. in which case what is the Ed relative to the MLD? as it is possible to still have significant quenching below the MLD if the Ed is deeper

Page 7, line 2: there is no mention in the methods section on quality control about positive spikes being removed from the bbp profiles. On the contrary in the methods section is says that the positive spikes are retained. Please describe the method used to remove positive spikes. Also, please clarify whether the positive spike data set was

retained separately so that flux estimates as per Briggs et al., could still be performed?

Page 10, line 10: I am not convinced that you can say anything concrete about the representativeness of the previous model without having a handle on the errors in the bio-argo chla data? if the errors in the chla data can be as much as 100% it is likely that they would significantly affect the shape of the 3 order polynomial fit. In particular all the SO data points which lie above the Morel fit are likely to "in reality" all be shifted to the left (i.e. lower chla) and closer to the Morel model line?

Page 10, line 20: I think that a really interesting discussion here would be the regional range in errors in chlorophyll associated with the global application of dividing the chla data by 2.

Page 11, line 1: The discussion does well to highlight the number of profiles, the regional coverage etc but I think what is lacking is a discussion of the benefits of a high resolution long term time series of biological and physical parameters that bio-argo can provide e.g. showing both seasonal and sub-seasonal variability….and in some cases perhaps even inter-annual variability (depending on the life time of the float or the succession of floats in a similar water mass). I would suggest that this ought to be highlighted with an example time series from one of the floats showing physics (e.g. temp) and biology (e.g. chlorophyll).

Page 11, line 15: I think that it would be good to mention some of the other bio-argo data bases that are currently available e.g. SOCCOM and perhaps plans to integrate them if any?

Table 2: Perhaps add the abbreviations to the Basin section of the Table to reflect those in Figure 1.

Figure 1. It is hard to see both the surf and prof stations on this Figure. I wonder if they would be clearer if you reduced the size of the blue dots slightly and outlined the red diamonds in black.

Please also note the supplement to this comment:
https://www.earth-syst-sci-data-discuss.net/essd-2017-58/essd-2017-58-RC2-supplement.pdf

---

## Author Comment (AC1) · 5 Oct 2017

**Revision of the manuscript**

Two databases derived from BGC-Argo floats measurements for biogeochemical and bio-optical applications at the global scale

by Organelli et al.

We would like to thank Dr. Sandy Thomalla and the anonymous Reviewer for their constructive comments and suggestions to the manuscript, and appreciation of efforts required for compiling the databases. Hereafter, the point-by-point response (**R#**) and action (**A#**) lists to Reviewer's comments **together with changes to the manuscript**:
* * *
**Reviewer #1**

The authors / data providers have prepared and presented a potentially useful data set. A user gets easy access and very good descriptions. Researchers have good options to use the data in depth profile formats or to take advantage of the euphotic zone summaries. The authors have performed a valuable service by compiling these particular bio-optical data from the larger ARGO data set and by specifying careful but consistent quality control procedures. The data seem like a very good fit to this journal. I applaud the effort and recommend publication. (Having Roesler et al. 2017 in open access proved very important for evaluation of this data set.)

I recommend changes to the presentation of this data that should make it much more useful. I make three over-arching suggestions followed by a sequence of line-specific technical suggestions or questions.

1) Global impact
The data set derives from 105 profilers operating over a time period of 40 months. Although one would have to dig deeply through the data to confirm, I believe that no single profiler operated for the entire 40 months. We have instead a compilation based on a series of relatively short (20 to 30 month) deployments each in a relatively restricted region. Although the map presented here as Figure 1 and the number of "stations" approaching 10'000 seem impressive, on the scale of an evolving global ocean they give us only a snapshot. A useful, unique, challenging snapshot, hardwon in the face of funding and operational constraints, but a relatively short Atlantic-focussed glimpse none-the-less. This data set basically misses most of the Indian and almost the entire Pacific Ocean and, not surprisingly, stays well away from ice-influenced regions. We should feel very well served to have these data! But we should not pretend that they provide us an encompassing view of an evolving global ocean. These authors hint at these limitations in their conclusions (page 11, line 15), where they mention "new regions", "improved vertical and temporal frequency", etc. Against these cautions, I feel that the title which includes the phrase "biogeochemical and bio-optical applications at the global scale" greatly overstates the impact.
**R1/A1:** The title has been modified to: "*Two databases derived from BGC-Argo float measurements for marine biogeochemical and bio-optical applications*". In the text, any reference to a "global" database has been removed.

The authors might consider several other efforts to compile global biologically-relevant or carbon cycle-relevant data sets for the oceans, including Peloquin et al. (HPLC chlorophyll) and Valente et al. (near-surface bio-optical properties) - both of which they do cite - or Sauzéde et al. (in situ fluorescence) or Bakker et al. (SOCAT, surface ocean CO2) which they do not cite. Taking only those four examples (all from ESSD), we typically see data collected over 2 to 6 decades presenting several 10s of thousands (up to millions, for SOCAT) of stations, profiles and measurements. In this paper we see nearly 10'000 observations in only 40 months (closer to 5'000 for the first optical depth) - the promising impact of ARGO technology which the authors could highlight more clearly - but also clear temporal and spatial limitations compared to other data compilations. In presenting their very careful quality control discussion, these authors have failed to show us clearly what their bio-ARGO profilers have achieved for ocean observations and also how these data contribute to, fit with, supplement, or surpass prior and on-going pan-oceanic data compilations. For this reader, these authors have missed an opportunity to quantify how "The BGC-Argo sampling approach can therefore help the scientific community accumulate observations on biological and biogeochemical properties of the ocean" (page 2, line 10). We want to do more than merely "accumulate"?
**R2:** Yes, we do not want merely to accumulate data. Data accumulation represents only the first step for

moving towards a more in-depth ecological, biogeochemical, and climatic understanding of the oceans. As we have mentioned in Introduction (2[nd] paragraph) and Conclusion (2[nd] paragraph) of previous (and revised) ms, thanks to such observations we can analyze particulate organic carbon fluxes (e.g., Dall'Olmo and Mork, 2014; Poteau et al., 2017), improve understanding on the phytoplankton phenology (e.g., Lacour et al., 2015) and impact of the physical forcing (e.g., Stanev et al., 2017) in a new and systematic way. BGC-Argo floats data are also useful to increase our understanding on how bio-optical properties of the oceans vary, and thus improving ocean-color remote sensing applications (e.g., Organelli et al., 2017; Barbieux et al., 2017). All these mentioned studies used as examples rely on BGC-Argo float data presented in this paper. It is worth to put particular attention on the studies by Organelli et al. (2017, JGR-Oceans) and Barbieux et al. (2017, JGR-Oceans) which directly exploited BOPAD-prof and BOPAD-surf as presented in this study. Hence, several efforts are in place to prove and advertise the scientific merit of such observations. We are, however, aware that this can be achieved only by involving the largest oceanographic community, and BOPAD-surf and BOPAD-prof should also serve to tempt and approach scientists to the BGC-Argo world. We think that specific scientific applications deserve to be addressed in more focused and specifically-designed studies.

A2: Page 2, Lines 9-14 (revised ms): the sentence has been rephrased in order to show that BGC-Argo measurements come to supplement already existing biogeochemically-relevant databases for the ocean: "*BGC-Argo can therefore help the scientific community to accumulate observations on biogeochemical properties from the surface to the interior of the ocean in a new and systematic way (Claustre et al., 2010a; Biogeochemical-Argo Planning Group, 2016; Johnson and Claustre, 2016). This, together with other several recent efforts to compile global biologically- or biogeochemically-relevant datasets (Peloquin et al., 2013; Sauzède et al., 2015; Bakker et al., 2016; Mouw et al., 2016; Valente et al., 2016), may provide new insights on marine ecological and biogeochemical processes and help understanding better if oceans and their properties have changed and/or are changing over the decades (Organelli et al., 2017).*" Bakker et al. (2014) and Mouw et al. (2016) have been added to the reference list.

Page 2, Lines 20-28 (revised ms): we have introduced specific examples that highlight the scientific merit of BGC-Argo floats in several fields of research: "*All these measurements, and derived quantities, are useful both for biogeochemical and bio-optical studies, to address the variability of biological processes (e.g. phytoplankton phenology and primary production; Lacour et al., 2015) and linkages with physical drivers (Boss et al., 2008; Boss and Behrenfeld, 2010; Lacour et al., 2017; Mignot et al., 2017; Stanev et al., 2017), to estimate particulate organic carbon concentrations and export (e.g. Bishop et al., 2002; Dall'Olmo and Mork, 2014; Poteau et al., 2017), and to support satellite missions through validation of bio-optical products retrieved from ocean color remote sensing (e.g. chlorophyll concentration; Claustre et al., 2010b; IOCCG, 2011, 2015; Gerbi et al., 2016; Haëntjens et al., 2017) or by identification of those regions with bio-optical behaviors departing from mean-statistical trends (i.e. bio-optical anomalies; Organelli et al., 2017).*" Bishop et al. (2002), Boss et al. (2008), Boss and Behrenfeld (2010), Poteau et al. (2017), Stanev et al. (2017), Lacour et al. (2017), Mignot et al. (2017) and Haëntjens et al. (2017) have been added to the reference list.

Page 14, Lines 10-21 (revised ms): we remind the reader of specific biogeochemically- and/or optically-relevant published studies based on the use of BGC-Argo floats included in this work, and that the way by which BGC-Argo floats collect data can supplement and complement existing pan-oceanic compilations: "*The two databases presented here can be directly exploited for several applications, from biogeochemistry and primary production estimation and modeling, to the analysis of the physical forcing on biology together with the assessment of any seasonal and sub-seasonal dependence, and to the evaluation of ocean's bio-optical variability. For specific examples based on same PROVOR-CTS4 profiling floats included in this study, the reader is referred to the works by Dall'Olmo and Mork (2014) and Poteau et al. (2017) for estimation and analysis of particulate organic carbon concentrations and fluxes; Lacour et al. (2017), Mignot et al. (2017) and Stanev et al. (2017) for observing physical impacts on biology; and Organelli et al. (2017) and Barbieux et al. (2017b) for analysis of the variability in diffuse attenuation coefficients for downward irradiance and particle optical backscattering-to-chlorophyll ratios across different oceanic areas, respectively. It is worth noting that the latter two studies have been pursued by exclusively exploiting BOPAD-surf and BOPAD-prof. The new and systematic way BGC-Argo floats collect data, and their potential in dramatically increasing oceanic observations in a restricted time, come also to supplement and complement published carbon cycle- and optically relevant pan-oceanic data compilations (Peloquin et al., 2013; Sauzède et al., 2015; Bakker et al., 2016; Mouw et al., 2016; Valente et al., 2016).*"

Please see **R16/A16** for comparison between the here presented and previously published databases.

2) Representativeness
From the understandable view of these biological and bio-optical oceanographers, the ideal ocean situation occurs when a profiler reaches the ocean surface near local noon with small waves and a cloud-free sky. In this paper we encounter a series of qualitative statements about variances from those ideal conditions:
Page 7, line 33 "unstable" meteorological conditions.
Page 8, line 4 - Again a focus on stability (now of the water column!) and "deteriorated sky and sea conditions".
Page 8, line 24 "worsening meteorological conditions and deepening mixed layer depths".

But, the global ocean represents a windy, cloudy, stormy place. Large regions have persistent coverage of stratus clouds at certain times of the year. For some parts of the ocean we have almost no cloud-free images despite nearly 40 years of daily satellite observations. From a biological view, we should appreciate disturbed conditions and vigorous mixing processes. I believe the standard (non-biological) ARGO profilers rise to the surface without regard to sky or wind? If, in this data set, the authors, consciously or sub-consciously, focus on and lead the user toward mid-day profiles under calm seas and clear sky conditions, we together suffer the risk of developing a serious bias?

**R3:** Qualitative statements Reviewer#1 mentioned only refer to radiometric profiles. Although the sampling strategy provides acquisition of all variables at solar noon regardless of sea and atmospheric conditions, quality-control procedures take meteorological conditions into account only for radiometric quantities. This emerges also from Figure 6, where the number of radiometric profiles is lower than for T, Chl and the other variables. Following which, the presented databases of Chl, FDOM and $b_{bp}(700)$ are not biased to calm seas and clear sky conditions.

Instead, the rationale of a more rigid quality-control for radiometry relies on the need of a perturbation-free (e.g., from moving clouds) profile in order to appropriately calculate quantities such as the euphotic depth and diffuse attenuation coefficients for downward irradiance ($K_d$). However, as discussed in Organelli et al. (2016a), this quality-control only verifies the shape of the profile, and accepts as good also those profiles collected under overcast sky as soon as this condition lasts all the cast. As a confirmation, Figure 11 shows comparison between in situ and satellite $K_d$ coefficients that represent 10% only of BOPAD-surf and correspond to bio-optical quantities derived for cloud-free measurements.

However, we are aware that radiometric profiles acquired under stormy, windy or cloudy conditions might be useful for applications such as primary production modeling. In this case, no quality-control other than on measured values is required and data can be directly downloaded from the data center at ftp://ftp.ifremer.fr/ifremer/argo/dac/coriolis.

**A3:** Page 5, Lines 9-11 (revised ms): the following sentence has been added: *"This protocol accepts as good measurements acquired both under clear and cloudy sky conditions, as soon as these remain stable during the cast (Organelli et al., 2016a)."*

Page 5, Lines 20-21 (revised ms): *"For radiometric data prior to the application of the above-mentioned quality-control procedures, the reader is referred to the archive at ftp://ftp.ifremer.fr/ifremer/argo/dac/coriolis."* has been added.

Page 8, Line 20 (revised ms): the sentence has been modified to *"between January and April. This occurs especially…"* to specifically refer to radiometric quantities.

Page 14, Lines 6-7 (revised ms): the following sentence has been added: *"…collected in just three years despite of meteorological conditions in several oceanic areas with depths greater than 1000 m"* to remind the reader that the databases are not biased to calm sea and clear sky conditions.

3) Accuracy

Having overcome many of the serious technical, operational and funding challenges of gaining useful bio-optic data from autonomous ocean profilers, and having applied a consistent set of quality control procedures intended primarily to remove spikes and outliers (and secondarily to identify instrument drift) the authors then present all data as uniformly certain. Although the authors show data distributions in several plots, we see no error bars and no uncertainty shading. For intercomparisons, these authors give us only global ranges (min-max values).

The description needs to thoroughly address uncertainty in a specific section. After all processing, what remaining uncertainties apply to what data? What changes in operation, instrumentation or data processing could or could not address those uncertainties? For the derived properties ZEU and ZPD we get statistical (standard deviation) uncertainties (e.g. Table 1 on page 17) but those derive from the data processing and do not address uncertainties in the underlying measurements? For many real-world measurements uncertainty remains very difficult to quantify but in this case the reader needs from the authors at least a sense of confidence and uncertainty for each product. If, as the authors clearly hope, these data prove useful for global models, quantitative uncertainty information will prove an absolute requirement.

**R4:** As Reviewer#1 stated, quantification of experimental uncertainties associated to each variable and assessment of the error budget is a very complex and long task. We agree, however, that readers need to be provided with at least a sense of confidence on the data included in BOPAD-prof and BOPAD-surf. Our group is already putting several efforts in improving qualification of each variable, and characterize and reduce uncertainties. Most of the studies are still underway (e.g., NPQ; radiometry temperature-dependence) but we have decided to anticipate their impact on the measurements anyway (for details please see **A4**). For variables such as the $b_{bp}(700)$, studies based on same sensors and/or platforms have been already published and so they can be used as a support to these databases (e.g., Dall'Olmo et al., 2009;

Briggs et al., 2011; Sullivan et al., 2013; Poteau et al., 2017). In the case of Chl, FDOM and $b_{bp}(700)$, we also performed a sensitivity analysis for three specific examples presented in the paper by calculating the impact of the sensor sensitivity threshold on the final quality-control product (see figure below). This analysis revealed that high impacts, thus errors, occur only in specific cases or in some part of the profile (e.g., for deep Chl values or surface FDOM concentrations). In the case of BOPAD-surf, standard errors were already associated to diffuse light attenuation coefficients and revealed to have a median impact of about 5% on derived products.

[Figure]

*Impact of the sensor's sensitivity limit on quality-controlled values for three specific examples of Chl, FDOM and $b_{bp}(700)$ profiles. Sensor's sensitivity limits are: 0.007 mg m$^{-3}$; 0.28 ppb of quinine sulphate and 2.2\*10$^{-6}$ m$^{-1}$ for Chla, FDOM and $b_{bp}(700)$ respectively.*

**A4:** Pages 11-14 (revised ms): We added Section 6 ("Data uncertainty") in which we describe and discuss uncertainties associated to each variable within BOPAD-prof and BOPAD-surf. The section reads: "

*Through this section, characterization of the uncertainty associated with each quality-controlled variable within BOPAD-prof, and for derived products contained in BOPAD-surf, is provided. No error propagation and budgets are here presented.*

*When using fluorescence measurements as a proxy of Chl concentration, the uncertainty may propagate from conversion of electronic counts in geophysical quantities, through the application of quality-control procedures for the influence of the non-photochemical quenching (NPQ) and/or other environmental variables (e.g., non-algal matter), to calibration corrections. The sensor sensitivity of 0.007 mg m$^{-3}$ (i.e. 1 digital count) is critical at the surface of most oligotrophic environments or for low Chl deep values, where it may be twice as high as the signal (Fig. 12a). Correction for the non-photochemical quenching may also introduce uncertainties depending on the procedure and assumptions on which the method relies on. However, a comparison between the method by Xing et al. (2012) used here and based on the calculation of the mixed layer depth, and an alternative correction developed by Sackmann et al. (2008) based on the use of particle optical backscattering, showed similar performances for BGC-Argo Chl measurements (Xing X., unpubl.). As discussed by Xing et al. (2017), the correction of Chl profiles for non-algal matter disturbance by using alternative procedures at the one applied here may also introduce errors, which vary regionally and are the highest in the Black Sea area (~0.1 mg m$^{-3}$) while the lowest are observed in the subpolar North Atlantic and Mediterranean Sea (~ 0.007 and 0.004 mg m$^{-3}$, respectively). A main challenge in quality-assessing fluorescence Chl measurements relies on the assumption between what is measured and what is actually phytoplankton biomass (Roesler et al., 2017). The fluorescence-to-chlorophyll ratio depends on changes in nutrient availability, growth phase, photophysiology and taxonomic composition of algal communities (Cullen, 1982). This implies that calibration factors may change regionally and seasonally. Indeed, standard fluorometer corrections relies on the comparison with contemporaneous HPLC-determined chlorophyll concentrations which are the most accurate estimates for phytoplankton pigments. However, given the BGC-Argo particularity of sampling autonomously, over long-periods and across different regions, any HPLC-based calibration performed at the time of the deployment may become invalid during the float's voyage. Haëntjens et al. (2017) recommend the use of radiometric data available on floats together with models (Xing et al., 2011) to systematically verify the calibration of the Chl fluorometer, and applied corrections, over time. In this study, no spatio-temporal variability of the fluorescence-to-chlorophyll ratio has been taken into account to correct BGC-Argo Chl measurements as well as no radiometry-based corrections have been used to avoid redundancy among variables and derived quantities (i.e. $K_d(490)$). Only the correction for the instrument-induced bias recommended by Roesler et al. (2017) has been applied, though it might be insufficient and so under-correct Chl values measured at high latitudes and especially in the Southern Ocean (Roesler et al., 2017). Chl profiles prior to the application of any quality-control procedures used here, including non-photochemical quenching and the recommended calibration factor by Roesler et al. (2017), are also archived in BOPAD-prof so that alternative chains of protocols can be applied at the user's discretion.*

*FDOM measurements within BOPAD-prof appeared very noisy even after quality-control and spike detection (see Sect. 3). However, using the profile in Fig. 4b as a specific example, the impact of the sensor sensitivity (0.28 ppb, ~ 1 digital count) on the measured values may be critical for surface measurements (Fig. 12b). Low FDOM values at the surface may be a result of the attenuation by other optically significant substances of the light fluoresced by the dissolved material (Downing et al., 2012), and/or be quenched as an effect of increasing temperature (Baker et al., 2005). No specific methods to BGC-Argo floats measurements are currently available to correct for the thermal fluorescence quenching properties, and it has been preferred to avoid implementation of published procedures (Wratas et al., 2011; Downing et al., 2012; Ryder et al., 2012) as they can be applied at the user's discretion.*

*Uncertainties related to the particulate optical backscattering, as acquired by WETLabs ECO sensors or instruments with similar or same technical and geometrical characteristics, have been discussed by already published studies (Dall'Olmo et al., 2009; Briggs et al., 2011; Sullivan et al., 2013; Poteau et al., 2017). Experimental errors may arise from multiple sources such as conversion and calibration coefficients (e.g., scaling factor and dark counts), instrument age and sensor responsivity to environmental factors such as temperature and light (Sullivan et al., 2013). The impact of the sensor sensitivity (2.2\*10$^{-6}$ m$^{-1}$) on the measured values is low (Fig. 12c). The combined uncertainty is generally less than 10% (Sullivan et al., 2013), but it may increase up to about 30% in most oligotrophic environments*

*(Dall'Olmo et al., 2009). In particular, the recent analysis by Poteau et al. (2017), that includes the same BGC-Argo floats used in this study, suggests that more consistent $b_{bp}(700)$ measurements would be achieved by taking into account a bias equal to $3.5*10^{-5}$ $m^{-1}$ due to changes in dark counts from the time of sensor's purchase to that of deployment. Disagreement between different sensor models measuring $b_{bp}(700)$ in the same areas may yield a bias up to 30% (Poteau et al., 2017).*

*Experimental uncertainties in radiometric profiles may arise from instrument tilt with respect to the vertical (maximum of ±10%, Leymarie E., unpubl.,) and sensor calibration (2-4 %, Hooker et al., 2002). The shading of the float's antenna and CTD head is negligible for $E_d(\lambda)$ sensor, except over a few degrees of sun's azimuth (direct shading, Leymarie E., unpubl.). The study by Briggs et al. (in prep.) on radiometers implemented on the PROVOR-CTS 4 BGC-Argo floats also evidences the dependency of sensor dark counts on ambient temperature. The uncertainty in factory dark measurements is the lowest near 20 °C (<0.01 μW cm$^{-2}$ nm$^{-1}$ for $E_d(\lambda)$; < 1.4 μmol quanta m$^{-2}$ s$^{-1}$ for PAR), for both $E_d(\lambda)$ and PAR. Highest errors occur when the radiometer operates near 0 °C, as the uncertainty grows up to about 0.06 μW cm$^{-2}$ nm$^{-1}$ for $E_d(490)$ and 2.6 μmol quanta m$^{-2}$ s$^{-1}$ for PAR. Similarly, higher uncertainties are also observed when radiometric measurements are acquired around 30 °C (~ 0.03 μW cm$^{-2}$ nm$^{-1}$ for $E_d(412)$ and $E_d(490)$; Briggs et al., in prep.) than near 20 °C. It is important to note, however, that dark offsets generally affect profiles at depth as the irradiance drops to 0, whilst their impact is less than 1 % for the highest values at the top of the ocean (Organelli et al., 2016a).*

*In BOPAD-surf, the standard error is associated to each value of diffuse attenuation coefficient for downward irradiance and PAR ($K_d(\lambda)$ and $K_d(PAR)$) as derived from the linear fit on log-quantities within the first optical depth $Z_{pd}$ (see Sect. 2.4). Errors can have an impact up to 33% on the measured coefficients, although the median value for the entire database is less than 5% regardless of the waveband with the minimum found for $K_d(380)$ (i.e. 3.4%). Because Chl, FDOM and $b_{bp}(700)$ represent the mean value of the profile within $Z_{pd}$, the standard deviations are archived in BOPAD-surf. The median value of the coefficient of variation (CV%; calculated as 100*(SD-to-Mean ratio)), for the entire database, is low for all the three variables and around 5% in the case of FDOM and $b_{bp}(700)$. The variability in Chl concentration is close to 0% as a consequence of the application of the method by Xing et al. (2012) that corrects the non-photochemical quenching by extrapolating the Chl value at the bottom of the mixed layer to the surface. More importantly, such a low variability in the observed variables suggests that they were homogenously distributed within the first optical depth as derived from PAR measurements, and that $Z_{pd}$ was similar or shallower than the mixed layer depth."*

The references Hooker et al. (2002), Sackmann et al. (2008), Dall'Olmo et al. (2009), Xing et al. (2011), Ryder et al. (2012) and Briggs et al. (in prep.) have been added to the reference list. The figure shown above is Figure 12 in the revised ms.

These authors need to explicitly address issues raised by Roesler et al. In its present form, this manuscript appears to have added those issues, and applied a uniform 2x correction, after the fact or at least late in the preparation process. Roesler et al attempted to exclude from their analysis exactly those mid-day, clear sky, high irradiation conditions that this compilation seems to favour - how does that mismatch affect the values presented here? Roesler et al - using many of these same data! - showed a very strong regional dependence of correction factors, indicating that a global uniform application of a 2x correction factor would in fact prove seriously wrong in almost all cases for almost all regions. Yet here we read about a simple 2x correction for chlorophyll values? Roesler et al. provided explicit regional- or biome-based correction factors that this paper should have considered?

**R5:** The paper by Roesler et al. (2017) has tried to attempt a global evaluation of the calibration factor for in vivo Chl fluorescence measurements using various approaches (from algal cultures, to HPLC pigment analysis and optical techniques), avoiding other sources of uncertainties such the non-photochemical quenching (i.e., daily profiles). Main finding is the recommendation of applying, at the user level, a factor 2 to improve the global accuracy of chlorophyll concentration estimates for the WET Labs ECO sensors and products derived from them. After application of the factor 2, regional and seasonal variability may remain so that under- and overestimations of Chl concentrations may occur for specific areas (e.g., Southern Ocean and Black Sea). However, the regional HPLC-based calibration factors provided in Roesler et al. (2017) are not comprehensive of all the regions and season encompassed by this study. Hence, explicit-regional corrections could be applied only for a small fraction of the database. In addition, we preferred not to use the float derived calibration coefficients presented in Roesler et al. (2017) as they were obtained by using a combined approach based on irradiance measurements and uncorrected Chl concentrations. Hence, such an approach may yield data circularity for bio-optical applications relying on the variability between, e.g., light attenuation and phytoplankton biomass. Finally, the factor 2 is only a scaling factor that can be easily replaced at the user's discretion.

**A5:** Page 12 Lines 12-28 (revised ms): the following sentences have been added in order to make the reader aware of the possible limitations and errors in applying a factor 2 for calibrating in vivo fluorescence Chl measurements collected by BGC-Argo floats: "*A main challenge in quality-assessing fluorescence Chl measurements relies on the assumption between what is measured and what is actually phytoplankton biomass (Roesler et al., 2017). The fluorescence-to-chlorophyll ratio depends on changes in nutrient availability, growth phase, photophysiology and taxonomic composition of algal communities (Cullen, 1982). This implies that calibration factors may change regionally and seasonally. Indeed, standard fluorometer corrections relies on the comparison with contemporaneous HPLC-determined chlorophyll concentrations which are the most accurate estimates for phytoplankton pigments. However, given the BGC-Argo particularity of sampling autonomously, over long-periods and across different regions, any HPLC-based calibration performed at the time of the deployment may become invalid during the float's voyage. Haëntjens et al. (2017) recommend the use of radiometric data available on floats together with models (Xing et al., 2011) to systematically verify the calibration of the Chl fluorometer, and applied corrections, over time. In this study, no spatio-temporal variability of the fluorescence-*

*to-chlorophyll ratio has been taken into account to correct BGC-Argo Chl measurements as well as no radiometry-based corrections have been used to avoid redundancy among variables and derived quantities (i.e. $K_d(490)$). Only the correction for the instrument-induced bias recommended by Roesler et al. (2017) has been applied, though it might be insufficient and so under-correct Chl values measured at high latitudes and especially in the Southern Ocean (Roesler et al., 2017). Chl profiles prior to the application of any quality-control procedures used here, including non-photochemical quenching and the recommended calibration factor by Roesler et al. (2017), are also archived in BOPAD-prof so that alternative chains of protocols can be applied at the user's discretion."*

In its present form, without an explicit discussion of uncertainty, this manuscript will fail to meet the needs and expectations of many potential users.

Specific comments as follows:

Data considerations

Why, at http://www.seanoe.org/data/00360/47142/ (corresponds to doi http://doi.org/10.17882/47142) do we find two versions, version 1 and version 2? Readme text in version 2 explains the differences, but the deletion of these 20-some profiles received no mention or justification in the text. Properly, a doi should point exclusively to a single version of any data. Second version should carry a second doi.

**R6:** There are two versions because V1 has been primarily used and archived in order to accomplish with AGU data archiving policy when publishing the paper by Organelli et al. (2017). The paper here revised was under preparation at that time and, before submission, we decided to remove 19 samples (0.3% of the database) because of less accurate values for $K_d(PAR)$ and $Z_{eu}$. This was not considered a major change and, as suggested by the data publisher itself, we decided to remove those data and make a second version available. Such a change does not affect in anyway the study by Organelli et al. (2017) which was based on 2847 simultaneous $K_d(380)$ and $K_d(490)$ measurements, and robust statistical analyses with removal of outliers.

**A6**: Justifications have been added in the Readme file and available at http://doi.org/10.17882/47142. Revised ms now specifies that V2 of BOPAD-surf is the one described (see section 8 on Data Availability).

We get profile data in separate parameter-based files (e.g. CHLA, CDOM), space delimited. We get the derived optical depth data in one file, properly comma delimited. (.csv). Why do we not get all files in .csv format?

**R7/A7:** Data formats have been harmonized. BOPAD-surf is now a space delimited/.txt file as those contained in BOPAD-prof. Because the number of profiles retained for each variable, and the number of depths for a given profile, can be different among variables (as a consequence of the quality-control), the creation of a unique file for BOPAD-prof could lack in practicality for the users. The files contained in BOPAD-prof have been tested with common programming languages (e.g., R) and they can be easily read by standard functions. Page 15, Lines 14-15 (revised ms): the following sentence has been added: "Files included in BOPAD-prof and BOPAD-surf can be read in table format by using standard functions of most common programming languages.".

Floats to breach surface around local noon. 100 floats times 100 profiles per each gives roughly 10000 profiles. At once per 10 days, 100 profiles would take 1000 days, not quite 3 years. Average profile interval must therefore approach more like 13-15 days? No profiler operates for the full 40 months, but wrong to specify an average of 10 days over that full time period. More properly an average of 10 days while operating?

**R8/A8:** Page 3, Line 20 (revised ms): added "*while operating*". See **R11/A11** for calculation of the average profile interval.

Page 2, Lines 16, 17: "nitrate concentrations" Has any ARGO biogeo float solved the NO3 challenge? Not sure why the authors mention this?

**R9:** A very few PROVOR-CTS4 profiling floats have been equipped with nitrate sensors (SUNA-v2). Quality-controlled data have been presented and made available by Pasqueron de Fommervault et al. 2015 (Journal of Geophysical Research, 120, doi:10.1002/2015JC011103).

**A9:** Page 2, Line 19 (revised ms): "*nitrate concentrations*" has been removed from revised ms because this paper does not deal with nitrate data.

Page 4, correcting the FDOM. Removed spikes outside 25 and 75, and then additionally remove spikes

greater than 4 x the mean value? Identification and quantification of temporal deterioration assumes consistency of deep water masses?

**R10:** As FDOM profiles are very noisy, spike removal is achieved in two following steps: 1) removal of points outside the 25 and 75 quantiles; 2) removal of any measurement with absolute residual value > 4 calculated as the difference between the profile and a mean filter. Identification of temporal deterioration does not assume consistency of deep water masses.

**A10:** Page 4 Lines 28-30 (revised ms): The procedure for removing FDOM spikes has been clarified as follows "*2) removal of negative and positive spikes outside the 25- and 75-quantiles of the raw profile, and subsequently purge of any measurement with absolute residual value > 4 calculated as the difference between the profile and a mean filter.*"

Please see **A37**, for changes to Section 2.3 on the identification of temporal deterioration.

We need to know the duration of each profiler's operation. Can we determine the average lifetime of each float? E.g float 7900591 operated from 2013-12-20 to 2015-07-05, e.g. 30 months, over which time it took roughly 70 profiles (average interval of 13 days but even greater because for the first month after deployment it apparently profiled once per day). We also need number of profiles per profiler? To understand anything about contamination or other performance deterioration as a function of cumulative time in the water, we need to know average deployment duration for the profilers as well as number of profiles by each. Not hard to extract and perhaps graph this information? This information would also prove helpful in making the points about large effort and great resources needed to achieve even this level of spatial and temporal coverage.

**R11/A11:** In Table S1 (Appendix A of revised ms) we have added for each float: date of first and last profile; lifetime (units of days); number of profiles and the average profile interval (units of days). The latter has been calculated as lifetime/number of profiles, although we are aware that frequency of a single profiler could have been modulated during the float lifetime in order to switch to an adaptive sampling able to resolve specific biological events such as phytoplankton blooms. We have also reported average values for the entire database.

Page 3, Line 21 (revised ms): "*(every 4±2 days on average, see Appendix A)*" has been added.

Page 5, line 2: "Ed (0-)" and Page 5, line 9: " Ed (0+ )" typos? How related to Ed (lambda )?

**R12:** This is the standard terminology used to indicate the irradiance just below (i.e., $E_d(0^-)$) or above (i.e., $E_d(0^+)$) the sea surface. As a general reference please see Mueller et al. (2003) cited in the ms. $E_d(0^-)$ is extrapolated from the vertical $E_d$ profile using a second-degree polynomial function (Organelli et al., 2016a). $E_d(0^+)$ is $E_d(0^-)$ corrected for the irradiance loss at the air-sea interface.

**A12:** Page 5, Lines 14-21 (revised ms): the sentence has been modified to: "*Because $E_d(\lambda)$ and PAR measurements are collected up to few centimeters from the sea surface, quality-controlled vertical profiles were completed by values just below it ($E_d(0^-)$). The $E_d(0^-)$ values were calculated by extrapolation within the first optical depth ($Z_{pd}$) using a second-degree polynomial fit (Organelli et al., 2016a), with $Z_{pd}$ calculated as $Z_{eu}/4.6$ (Morel, 1988). The euphotic depth, $Z_{eu}$, is the depth at which PAR is reduced to 1% of its value just below the surface and was calculated from measured PAR profiles. To achieve $E_d(0^-)$ calculations, an initial value of $Z_{eu}$ and $Z_{pd}$ were firstly estimated from the shallowest PAR measurement and subsequently from that corresponding to $0^-$.*"

Page 6, Lines 4-9 (revised ms): the sentence has been modified to: "*Test 4 was based on the comparison between irradiance values just above the sea surface ($E_d(0^+)$) with those modeled by Gregg and Carder (1990) for clear cloudless sky, as described by Organelli et al. (2016a). The performance of this test, which assesses the accuracy of measured irradiance values, strongly depends on the value extrapolated to the sea surface (i.e. $E_d(0^-)$). $E_d(0^+)$ values at 380, 412 and 490 nm were obtained by dividing $E_d(0^-)$ derived from quality controlled profiles as described in Sect. 2.2 by the transmission across the sea–air interface factor (Austin, 1974).*"

Austin (1974) has been added in the reference section.

Page 5, line 17, tracking possible biofouling. Interruption of the time series occurred in real-time or during post-processing?

**R13/A13:** Page 6, Line 10 (revised ms): "*pre-processing*" has been now specified.

Page 6, line 2 - simple vertical average of CHLa, CDOM, etc., for optical depth? How did these optical depths compare to other data, globally or regionally?

**R14:** Yes, it is a simple average within the first optical depth. This is similar to the procedure used by Valente et al. (2016; end of page 237). Standard deviation of the mean has been included within BOPAD-surf.

**A14:** none.

Page 6, comparison with satellite data.
**R15/A15:** corrected.

The authors have missed an important opportunity to connect and compare these data to the bio-optic data reported by Valente et al. These authors cite that data set (page 9, line 20), but only once for the purpose of confirming the bio-optic properties reported here (and buried in a summary sentence in which the reader can't determine which external paper connected to which parameter reported here). In fact the Valente et al data represent an important partner for these data. Those data stop at 2012, these data extend through 2015. This data includes Labrador Sea and Southern Ocean locations missing in the Valente et al. data. That data has many more values, e.g. for Chl A, that could give a much better regional comparison (North Atlantic, for example) for these data. These authors do not need to do or show the work of attempting to merge this data into a Valente et al. framework but they should explicitly outline the connection points and opportunities. These authors could also very much learn from the graphic approaches (e.g. Valente et al. figure 3 and figure 10) and users of both data sets need some convergence of variable names and units (e.g. for FDOM treated very differently in the two data sets).

**R16:** Yes, BOPAD-surf and Valente's database may be good partners. The whole community would benefit of the comparison highlighting connections, complements and differences. As Reviewer#1 outlines, BOPAD-surf has shorter temporal coverage than Valente's database but consecutive. It includes data from high latitudes while Valente includes the North Pacific and Indian oceans areas where no BGC-Argo "PROVOR-CTS4" floats have been deployed. Moreover, BGC-Argo floats can complement the database by Valente by providing variables, in both hemispheres, also for wintertime and other harsh sea periods (please see Fig. 3 and related comments in Valente et al., 2016). Considering parameters, only two variables are directly comparable between the 2 databases: $K_d(412)$ and $K_d(490)$. BOPAD-surf contains also $K_d(PAR)$ and $K_d(380)$, with the latter especially important in view of upcoming new satellite missions (e.g., ESA's Sentinel 3 and NASA's PACE). BGC-Argo can complement the spectral resolution of particle backscattering measurement by Valente, as we provide such data at 700 nm (Valente is up to 683 nm). Valente et al. (2016) deliberately removed all Chl concentrations derived from in vivo fluorescence measurements due to the actual calibration issues we encountered with BGC-Argo and that both Reviewers highlighted. As Reviewer#1 suggests, in the future, Valente et al. may provide an additional help to our efforts in getting the best calibrated Chl from in vivo fluorescence Argo measurements. Regarding measurements of colored dissolved organic material (CDOM or CDM if we include the detrital particulate matter), the FDOM used in BOPAD-surf is a different parameter from $a_{dg}$ used in Valente. While $a_{dg}$ relies on the light absorption of the whole pool of CDOM, FDOM is a measure of fluorescence and, depending on the excitation/emission wavelengths of the sensor, represents a measure of the fraction of freshly-produced or more humic and refractory matter. In any case, FDOM is only a fraction of the entire CDOM pool and their ratios seem to be highly varying region-by-region and temporally within a given area. The community is, however, far from a comprehensive understanding of this variability both at the regional and global scale. Though $a_{dg}$ is the parameter directly retrievable from Ocean Colour Remote Sensing, FDOM represents useful data resource as well to in situ understand the optical behavior of the oceans (please see Organelli et al., 2017 as an application). Because FDOM and $a_{dg}$ are based on different measurement principles, treatments and units are different and cannot be harmonized.

**A16:** Page 11, Lines 3-28 (revised ms): the following paragraph has been added: "*BOPAD-surf does not, however, represent the only effort in compiling an extensive database tailored for in situ and remote bio-optical applications. BOPAD-surf and the compilation published by Valente et al. (2016; hereafter VL2016) may be good partners, and thus be beneficial to the largest community of oceanographers as soon as complementarities and differences are highlighted. Though BOPAD-surf's temporal coverage is shorter than for VL2016, it extends bio-optical measurements through 2013-2015. It includes regions such as the North Atlantic subpolar gyre, the Southern Ocean and the Red Sea that are not archived in VL2016. On the contrary, VL2016 offers data from the North Pacific and Indian oceans where no "PROVOR-CTS 4" profiling floats have been deployed (Fig. 1). BOPAD-surf complements VL2016 by providing a balanced acquisition of variables also during wintertime and harsh periods. Considering variables and differences in acquisition and processing, only the diffuse attenuation coefficients for downward irradiance at 412 and 490 nm (i.e. $K_d(412)$ and $K_d(490)$) are directly comparable between the two databases. VL2016 offers a 25-band resolution of these coefficients in the visible range, while BOPAD-surf extends such a measurement to a single wavelength in the UV region (i.e. $K_d(380)$) and includes attenuation coefficients for one broad waveband (i.e. $K_d(PAR)$). Similarly, BOPAD-surf provides measurements of the particulate optical backscattering at 700 nm, a band not included in VL2016 (27 bands between 405 and 683 nm). The main differences between the two databases appear for the variables relating to Chl and colored dissolved or detrital material (FDOM). Because of calibration challenges for deriving accurate Chl concentrations from in vivo fluorescence measurements (see Sect. 6), VL2016 is only compiled with Chl concentrations obtained from High Performance Liquid Chromatography (HPLC) and/or spectrophotometric/fluorometric measurements on algal pigment extracts. FDOM is a different parameter from $a_{dg}(\lambda)$ in*

*Valente et al. (2016). While $a_{dg}(\lambda)$ relies on the light absorption properties of the whole pool of colored dissolved and/or particulate organic material, FDOM only measures the fluorescence emitted by a fraction of this matter. Depending on the excitation/emission wavelengths of the sensor, FDOM can be a proxy of concentrations of freshly-produced material or more aged humic substances (Nelson and Gauglitz, 2016). However, in some regions, FDOM can be significantly correlated to $a_{dg}(\lambda)$ and thus retrievable from ocean color remote sensing (e.g., Matsuoka et al., 2017). FDOM data included in BOPAD-surf also represent a useful resource to improve the understanding on the optical behavior of the oceans (Organelli et al., 2017). Finally, no measurements of remote sensing reflectance are archived within BOPAD-surf, but successors of the PROVOR-CTS 4 profiling floats used in BOPAD-surf are planned to be deployed in order to collect multispectral downward irradiance and upwelling radiance measurements.*"

Nelson and Gauglitz (2016) and Matsuoka et al. (2017) have been added to the reference list.

Page 7, line 22 "open-ocean environments" If by "open-ocean" the authors mean 'collected in areas with depths greater than 1000 meters, as opposed to shallower continental shelf regions, then we can perhaps accept their definition so long as they describe what they mean more carefully. If, however, "open-ocean" should imply a broad spatial coverage of large ocean regions, then the absence of measurements from the Pacific stands out. The authors can correctly say `within limitations of project-driven resources, deployments focused on some of the important carbon-export regions of the Atlantic Ocean, in all cases in regions with depths greater than 1000 meters'?

**R17/A17:** Page 8, Lines 4-7 (revised ms): the sentence has been rephrased as: "*Deployment of BGC-Argo floats has been mainly focused, within limitations of project-driven resources, on some of the important carbon-export regions of the Atlantic Ocean (Alkire et al., 2012), on areas with dynamic trophic regimes (e.g. Mediterranean Sea; D'Ortenzio and Ribera d'Alcalà, 2009) and on oligotrophic mid-ocean gyres, in all cases in regions with depths greater than 1000 m.*". Sarmiento et al. (1992) and Takahashi et al. (2002) have been removed.

Page 8, line 31. And North Pacific equals 0 %.
**R18/A18:** Page 9, Lines 17-18: the following sentence has been added: "*Large areas such as the North Pacific and Indian oceans equal 0 % as no deployments occurred in those regions.*"

Page 9, line 16. I don't believe, from this data set, the authors can make any statistically-valid statement with respect to any part of the Pacific.
**R19:** The sentence only describes results of Figure 9. No statistically-valid statement was intended.
**A19:** None.

Page 9, final paragraph. Again, a good description of percentages within regions of deployments, but must include recognition of very large areas (e.g. Pacific) with no deployments.
**R20/A20:** Page 10, Lines 23-24 (revised ms): the following sentence has been added: "*North Pacific and Indian oceans equal 0%.*"

Page 19, legend to Figure 1. Dot and diamonds hard to distinguish, should specify red diamonds and blue dots.
**R21/A21:** done.

Page 19, Figure 2. Reader to assume that data from float 6901439 were truncated at some point, with data after that time point discarded?
**R22:** Yes, the point from which data have been disregarded is between 2 and 2.5 ppb of quinine sulfate.
**A22:** Please **R50/A50**.

Page 20, Figure 3. Panel A - Why assume only non-photochemical quenching? Does NPQ include by definition active surface avoidance? Panel C includes the 2x factor?
**R23:** It assumes NPQ because Chl is supposed to be homogenously distributed within the mixed layer. So, any decrease in Chl concentration is mainly an effect of NPQ. We are sorry but we are not sure to have fully understood what Reviewer#1 means with "active surface avoidance". Panel C includes the 2X factor as it was highlighted at Page 6, Line 24 of previous ms.
**A23:** None.

Page 20, Figure 4. Why the blue circles indicate spikes in corrected (red) curves? Also, in this case, we assume corrected greater than raw due to a sensor performance issue tracked from the deep FDOM? But at what point would biofouling or sensor deterioration have disqualified the measurements?
**R24:** Spikes are indicated for uncorrected blue curves, what Reviewer#1 pointed out is only a result of the

graphics representation. Yes, the sensor can have small oscillations in the deepest values so that for two consecutive profiles we may adjust both to lower and great values. The measure would be disqualified if the entity of the correction becomes greater and greater over the time. In the specific case of Figure 4, both profiles have been collected within 1 week from the deployment.

**A24:** Caption of Figure 4: "*Cyan open circles indicate positive spikes for the raw profile. Both profiles have been acquired within one week from the deployment.*" has been specified. Figure 4 has been modified.

Page 21, Figure 5. My old eyes see blurred blue dots, no blue open circles.
**R25/A25:** "*blue open circles*" replaced with "*cyan open circles*", and circles have been made larger.

Page 21, Figure 6. Ascent speed, profile time? Variability of cloud shading over that time period?
**R26:** Because the speed of a float is nominally 10 cm/sec, the ascent 0-250 m $E_d$ profile is performed in about 42 minutes (2.8 h for 0-1000 m profiles). The major cloud shading characterizing the profile in Figure 6b is between 24 and 38 m, and corresponds to about 2.3 minutes of measurements. Assuming data acquisition every 1 m, the cloud influences at least 10 recorded data.
**A26:** Page 7, Line 31 (revised ms): the following sentence has been added: "*with the major cloud perturbing data acquisition for at least 2 min*".

Page 23, Figure 8 - Not a useful way to show geographic data, confusing. Use maps instead, as in Figure 9 (or Figure 10 of Valente et al.)
**R27:** According to Reviewer#1 suggestions, we have actually tried to re-draw the figure following the approach by Valente et al. (2016), but with no satisfactory results. In addition, such a modification would require a map for each variable thus largely increasing the amount of figures (from 11 to 19).
**A27:** none.

Supplement consists of a single table defining the acronym codes for specific ocean regions. If, as I suspect, it applies to or derives from a larger ocean regional description scheme, the authors should cite the external references. If it represents a product custom to ARGO generally or bio-ARGO specifically, the authors should include the table as an appendix in this ESSD manuscript. Otherwise, according to Copernicus archive procedures as this reviewer understand them, the archive process could preserve the manuscript but not the supplement.
**R28:** Supplement is a specific product to these databases. However, it is important to note that the studies by Organelli et al. (2017) and Barbieux et al. (2017), both based on BOPAD-prof and BOPAD-surf as presented here, use same repartition among regions/floats/profile number. They do not show float lifetime and the average profile interval.
**A28:** Supplement 1 is Appendix A of revised ms.
* * *
**Reviewer #2: Dr. Sandy Thomalla**

The processes and dynamics that define the climate sensitivity of the biological carbon pump are not well understood. This is due in part to our lack of understanding of this complex problem through chronic under sampling of the world's oceans, which do not resolve inter-annual variability and seasonal and intra-seasonal dynamics. Autonomous technology promises to overcome the space-time gap in ocean observations with bio-optical sensors on platforms that are able to profile the water column providing highly cost-effective measurements at high frequency that can characterise the vertical biogeochemistry at smaller scales, but also for sufficiently long periods that may help to reduce uncertainties associated with carbon budgets at longer time scales. As such, I recommend this highly useful data set for publication and commend the efforts of the authors in all the steps that such an achievement requires; from securing the funds to purchase the numerous floats to arranging for their deployment in a globally diverse manner all the way through to the significant efforts in processing and collating the data into a succinct repository.
However, although I see very obvious benefits in the use of such a database both for ocean colour product validation and to further our understanding of ecosystem dynamics, I have one major concern with regards to utilising the chlorophyll (chla) data for validating ocean colour. The uncertainties in the BGC-Argo chla data are typically large and poorly characterised – often larger than the satellite derived chla estimates (mainly due to the globally applied factor of two bias in the conversion of fluorescence to chl and the simple quenching correction which is difficult to evaluate without night time profiles). This raises some serious concerns with the use of float derived chl a data for match-up based validation application with

regard to uncertainty budgets. That being said however, I do not have a problem with the use of the other bio-opticalvariables (e.g. Kd, bbp, Zeu) for ocean colour validation, which are not susceptible to the same kinds of mismatches in the uncertainty budgets.

**R29/A29:** Please see from **R31/A31** to **R34/A34** regarding Reviewer's major concerns.

Inline with the above, I would recommend some changes to the manuscript that need to be addressed before being suitable for publication and provide some suggestions to improve the database. In addition, I provide a list of minor corrections and suggestions to improve the manuscript and attach a pdf with detailed comments and typos.

**R30:** Typos highlighted within the pdf file have been changed according to suggestions. Hereafter, replies to specific comments and questions in the pdf:

**A30:** Page 4 Line 7 (previous ms), Page 4 Lines 14-16 (revised ms): the sentence has been rephrased as: "*3) removal of measured values outside the specific range reported in the manufacturer's technical specifications (WETLabs, 2016). No interpolation of missing data was performed;*".

Page 4 Line 9 (previous ms), Page 4 Lines 16-17 (revised ms): the sentence has been expanded as: "*4) corrected for non-photochemical quenching (NPQ; Kiefer, 1973) by extrapolation of the maximum fluorescence within the mixed layer to the surface following Xing et al. (2012) and Schmechtig et al. (2014).*".

Page 4 Line 12 (previous ms): the sentence has been removed.

Page 4 Line 29 (previous ms), Page 5 Line 8 (revised ms): "*Positive spikes were retained.*" has been added.

Page 4 Line 18 (previous ms), Page 4 Lines 28-30 (revised ms): the sentence has been rephrased as: "*2) removal of negative and positive spikes outside the 25- and 75-quantiles of the raw profile, and subsequently purge of any measurement with absolute residual value > 4 calculated as the difference between the profile and a mean filter.*"

Page 5 Line 31 (previous ms), Page 6 Line 21 (revised ms): the sentence has been modified to: "*FDOM quality-controlled profiles were smoothed by...*"

Page 6, Line 27 (previous ms), Page 4, Lines 30-31 (revised ms): moved to "Data uncertainty" section.

Page 10 Line 3 (previous ms), Page 2 Lines 25-28 (revised ms): the following sentence has been added in Introduction: "*and to support satellite missions through validation of bio-optical products retrieved from ocean color remote sensing (e.g. chlorophyll concentration; Claustre et al., 2010b; IOCCG, 2011, 2015; Gerbi et al., 2016; Haëntjens et al., 2017) or by identification of those regions with bio-optical behaviors departing from mean-statistical trends (i.e. bio-optical anomalies; Organelli et al., 2017).*"

Page 10 Line 25 (previous ms), Page 10 Line 32 (revised ms): the following sentence has been added: "*plus understanding on the associated uncertainty*".

1. Major comments

1.1. Using BGC-Argo chla data for ocean colour validation

Although none of the BGC-Argo chla versus satellite chla matchups are presented in the manuscript, the implications to do so for validation purposes are implicit both in the text. (e.g. pg 3 line 4: "data presented in BOPAD-surf are compared with existing bio-optical models and used in conjunction with products derived from satellite platforms in order to show applicability for validating ocean-color bio-optical products at the global scale" and pg 9 line 5: "measurements collected by BGC-Argo floats are a fruitful resource of data for bio-optical applications . . ... as well as the validation of ocean color reflectance (Gerbi et al., 2016) and bio-optical products (IOCCG, 2015)" and pg 10 line 25 ". . .ocean-color algorithm and product validation can routinely be performed in several regions so that errors and possible causes of failure . . .. can be assessed and/or solved, and algorithms be refined for improving the quality of retrievals." ) and even more so in the data base itself (see http://seasiderendezvous.fr/matchup.php) where chla is the default product for match up locations and the colours of the data points represent the % relative error between float and satellite chla matchups. Given the inaccuracies in the float chla data I am not convinced that such a comparison is meaningful, in particular without any indication of the errors implicit in the BGC-Argo chla data. That being said however, I do not feel that the inaccuracies in the chla data render them ineffectual, on the contrary, these data will provide extremely useful information towards an improved understanding of the biological response to physical drivers and our understanding of the sensitivity of the biological carbon cycle to climate change that will ultimately lead to improved estimates of long term trends. For example, although the Southern Ocean bias in satellite estimates of chlorophyll is well known it does not render the data any less useful, it is however important that the user is well aware of the quantitative limitations of the data.

**R31:** Yes, the use of BOPAD-surf for validation of OC Chl products is implicit as it is for all OC products other than $K_d(490)$ shown in Figure 11. We acknowledge that error characterization needs to be better addressed (standard errors are included in BOPAD-surf; Section 8 revised ms and at

http://doi.org/10.17882/47142), but we do not think that such a comparison is not meaningful. BGC-Argo is a quite new technology, and procedures to quality-control data and define error budgets need time and resources (as it is for any in situ and remote measurement). This does not exclude a priori the potential of such platforms for OC applications. Indeed, efforts are underway to improve NPQ and calibration correction of Chl concentrations derived from fluorescence measurements.

**A31:** We have added a section on measurement uncertainty in the revised ms. Please see **R4/A4**.

From my understanding, one of the primary drivers of the errors in BGC-Argo chla is the variable relationship between fluorescence and chla which is not accounted for in the quality control step that divides all chla data by a factor of two to correct for the global bias in the factory calibration. Although Roesler et al., 2017 recommends to do so in order to improve the global accuracy of chla measurements from WET Labs ECO sensors, they acknowledge the regional variability in this factor, which ranges from 0.56 in the Arabian Sea to 7.75 in the Southern Ocean. As such, the global application of a factor of two can create errors that range from an underestimate of "actual" chlorophyll by ±100% in the Arabian Sea to an overestimate of chlorophyll in the Southern Ocean by ±250%. Would it not be possible to use some of the regional variability evident in the relationship between chla from HPLC and ECO-fl (Roesler et al., 2017, their Figure 1) to derive a more regionally robust factor for correcting the factory calibration bias?

**R32/A32:** please see **R5/A5.**

1.2. Quenching correction

Another area that can introduce a significant amount of error into both the profile and the surface chla data is the choice of quenching correction that is applied. The Xing et al., 2012 method of correcting quenching is robust and effective, so long as the assumptions it relies on are valid. The Xing et al. (2012), method relies on the assumption that a) chlorophyll concentrations within the mixed layer are uniform and b) that quenching processes do not affect depths below the depth of maximum fluorescence within the MLD. This method does not allow for sub surface fluorescence maxima to occur within the mixed layer. The method of Biermann et al. (2015) attempts to overcome this limitation by instead finding the maximum fluorescence within the euphotic layer and extrapolating this value to the surface. A comparison of their method with that of Xing et al. (2012) identified occasions (when the MLD was deeper than the euphotic depth) where quenching was corrected without masking subsurface fluorescence signals. However, as with the method of Xing et al. (2012), this method assumes homogeneity, but in this instance within the euphotic zone as opposed to the mixed layer (i.e. it does not allow for daytime subsurface maxima to be present within the euphotic zone). When these assumptions are not met (i.e. chlorophyll is not homogenous within either the mixed layer or the euphotic layer and quenching occurs below the mixed layer) the result will be a typical underestimate of daytime surface chla in the case of Xing et al. (2012) and an over correction of surface chla when the Biermann et al. (2015) method is applied. As mentioned in Xing et al, 2012, for multiinstrumented platforms with both fluorometers and backscattering sensors, Sackmann et al. (2008) proposed an elegant method that made use of the backscattering profile (as independent proxies of phytoplankton distribution) to correct the fluorometric one.

However, this method still relies on certain assumptions such as a regular association between particulate backscattering and chlorophyll concentration. Regardless, if both backscatter and fluorescence sensors are available then methods that utilise both parameters are perhaps more likely to retrieve accurate estimates of chlorophyll during the day.

Either way, it seems to me that there are at least two other methods of correcting quenching which ought to be applied to your data set and the results compared to try to determine which is the best method to use and when. Or at least have an idea of the different surface chla concentrations that the different methods produce in order to get a handle on the possible range of error that this quality control step can introduce. A major problem with having all the profiles in the data set being performed at midday (apart from the obvious issues with quenching) is that it is very difficult to quantify whether or not a daytime profile has been corrected correctly. As such, I would recommend that future float missions consider doing both midnight and midday profiles in order to improve the quality of the chla data (even if this means a reduction in the longevity of the float life span).

**R33:** Though being aware of possible limitations of the NPQ correction proposed by Xing et al. (2012) and highlighted by the Reviewer, this method has been applied after an appropriate analysis of benefits and comparison with alternative methods. One major advantage of the Xing method is that it can be applied to any dataset of Chl fluorescence (FChl) provided that the Mixed Layer Depth is known. It can therefore easily be executed on profiles acquired by CTD, gliders, sea elephants as well as by other configurations of

BGC-Argo floats (e,g., Xing et al., 2014 JGR-Oceans). As a result of this flexibility, the Chl profiles presented in BOPAD-prof may help building larger datasets of FChl beyond the BGC-Argo only. As a specific example, the Reviewer is referred to the paper/database published by Sauzède et al. (2015, ESSD). The NPQ correction proposed by Biermann et al. (2015) requires ocean color data that are not always available. In addition, Chl is extrapolated within the euphotic zone so that any increase of Chl with depth in stratified waters (due to light limitation and photoacclimation effect) could not be appropriately taken into account.

The method by Sackmann et al. (2008) uses $b_{bp}$ to shape FChl within the mixed layer. As a consequence, BGC-Argo profiles corrected with such a method could not be integrated within larger FChl database where the optical backscattering is not systematically acquired (e.g., Sauzède et al., 2015, ESSD). In addition, application of this method might result in circularity among variables especially if studies on the relationships between chlorophyll and particle optical properties such as $b_{bp}$ are attempted (e.g., Barbieux et al., 2017, in revision). Furthermore, an unpublished study by Xing X. shows that performances of Sackmann and Xing methods are similar with errors around 12%. Such a comparison is shown in the figure below where BGC-Argo profiles within the MLD collected at sunrise (when NPQ is assumed to be negligible) and at noon, before and after correction, are presented together with related statistics (such data are available only for a very small number of floats/profiles).

Finally, as highlighted in Section 7 of the previous version of the ms (now Section 8), uncorrected Chl profiles are also stored in BOPAD-prof so that different NPQ corrections can be applied at the user's discretion.

**A33:** Section 7 (revised ms): we make the reader aware about performances of alternative methods. Please see also **R4/A4** for changes in the text.

[Figure]

*Comparison between NPQ corrected FChl (mg m$^{-3}$) within the MLD measured by BGC-Argo floats at noon (y-axis) and at sunrise when NPQ is negligible (x-axis): Left panel) NPQ corrected using the Xing et al. (2012) method; Right panel) NPQ corrected using the Sackmann et al. (2008). In each plot, "v4" indicate the used NPQ depth criterion (i.e., min(MLD0.03, ziPAR20)). Red points indicate comparison before NPQ correction of the noon profile (MAE=0.2514, SMAPE=25.14%)*

1.3. Error estimates

Given that validation is a quantitative assessment of uncertainty and that the BOPADsurf data set is intended to be used for satellite validation, I feel that it is important to provide some indication of the anticipated errors in the derived variables. If, in the case of chl, you are wanting to validate a satellite product to within _35% uncertainty (in Case 1 waters) then it is important to know when your in situ product has an error of >100%. A short quantitative analysis of the expected uncertainties in the float derived chla data would be very useful and is necessary here. Similarly, I think that a more open discussion is required around the limitations and weaknesses of the published database together with its strengths.

**R34/A34:** Please see **R4/A4.**

2. Minor comments

Page 4 line 5: What about positive spikes in chla? I appreciate that they were retained as they could represent "real" data. However they appear to have been removed from the BOPAD-surf data base. If so please provide details. Also, if the spikes were remove, were they interpolated over in the vertical or left as NaN's?

**R35:** Positive spikes where retained in BOPAD-prof as they could bring useful biogeochemical information. This is also shown in Figure 3a. Chl concentration in BOPAD-surf represented the average value within the first optical depth. If not corrected by the NPQ correction, positive spikes were therefore retained. No interpolation of missing data was performed when removing negative spikes.

**A35:** Page 4, Line 14 (revised ms): "*No interpolation of missing data was performed. Positive spikes were retained*" has been added.

Page 4 line 10: is it possible to please clarify how you systematically determined the profiles that were affected by non algal fluorescence with depth? e.g. an increase in chla with depth for how many meters beyond what threshold depth? how do you ensure that you are correcting for non-algal increases of fluorescence with depth and not "real" subsurface increases in chla e.g. via a subducting water mass?

**R36:** According to procedures in Xing et al. (2017), the method was applied only when FChl and FDOM were linearly correlated below a given depth (please see quantitative metrics in Xing et al., 2017). The depth from which the linear fit was calculated has been determined profile-by-profile using Eq. 6 in Xing et al. (2017), then the fit is drawn up to 1000 m. The origin of this deep source of red fluorescence has been widely discussed in Xing et al. (2017) in conjunction with several published works on the topic. Microbial activity seems to be the main cause for such an occurrence, and anoxic and oxygen minimum zone of subtropical regions appear the most concerned areas by this feature. Though FChl indicating biological activity has been observed at 1000m depths in convection areas such as the Labrador Sea, Xing et al. (2017) showed that errors introduced by this correction are very low and on average around 0.007 mg m$^{-3}$ for such regions.

**A36:** Page 4, Lines 19-22 (revised ms): the sentence has been changed to: "*Profiles collected in areas such as the Black Sea and subtropical regions were further corrected for the contribution of fluorescence originating from non-algal matter following procedures described in Xing et al. (2017). The correction was applied when Chl and FDOM concentrations were positively correlated below the depth where Chl was supposed to be zero (for equations and quantitative metrics see Xing et al., 2017).*"

Page 5, line 11, step 3: this step is not clear to me? it has already been implied that positive spikes in some data were retained (e.g. chl and bbp) as they can represent "real" information. As such it is not clear to me how you used sharp gradients with depth to test for instrument drift?

**R37:** Sharps gradients are intended as a sudden increase/decrease of values for a given variable that cannot be explained with physical and biological phenomena. As it can be observed in the figure below, the increase of $b_{bp}(700)$ occurs simultaneously for the entire profile and it cannot be related to positive spikes (i.e., particle aggregates) as they are generally more sporadic over depth. In the specific example, the sensor also stops operating few cycles after the sudden increase. Such an occurrence can indicate both biofouling and instrument drift issues that, in the case of BGC-Argo floats, are hard to distinguish.

**A37:** The whole Section 2.3. has been restructured in order to clarify the methodology and where the analysis has been conducted on pre- or post-processing data. Section 2.3 now looks as: "*A set of four tests was specifically developed to identify potential biofouling and instrument drift. To achieve a reliable evaluation for each of the 105 BGC-Argo floats, each variable was examined both individually and in conjunction with the others, which is greatly aided by redundancy among derived quantities. A combination of raw profiles and quality-controlled products was needed for the analysis. Ancillary data such as measurements acquired in drift mode at 1000 m (i.e. between two following ascent profiles) were also included in the analysis and they can be publicly accessed at http://www.oao.obs-vlfr.fr/maps/en/. Test 1 was conducted on raw time-series of salinity, Chl, FDOM, $b_{bp}(700)$ and $E_d(\lambda)$, i.e, before the application of the quality-control procedures described in Sect 2.2. It aimed to identify sharp gradients in measured variables over the entire profile (i.e. sudden decrease/increase of Chl and FDOM concentrations or increase in $b_{bp}(700)$ values) not attributable to any biological or hydrological cause (e.g. particle aggregates or nepheloid layer of particles). Tests 2 and 3 were conducted on raw measurements collected by each profiler when in drift mode. Test 2 analyzed time-series of sensor's dark measurements for Chl and $E_d(\lambda)$ at the 1000 m parking depth. Test 3 consisted in the analysis of the relationship between raw FDOM and salinity at the 1000 m parking-depth over time. Assuming that deep CDOM concentrations are conservative in the same water body (Nelson et al., 2010), variations in deep FDOM values for a given salinity is likely due to changes in sensor performances (Fig. 2). Test 4 was based on the comparison between irradiance values just above the sea surface ($E_d(0^+)$) with those modeled by Gregg and Carder (1990) for clear cloudless sky, as described by Organelli et al. (2016a). The performance of this test, which assesses the accuracy of measured irradiance values, strongly depends on the value extrapolated to the sea surface (i.e. $E_d(0^-)$). $E_d(0^+)$ values at 380, 412 and 490 nm were obtained by dividing $E_d(0^-)$ derived from quality controlled profiles as described in Sect. 2.2 by the transmission across the sea–air interface factor (Austin, 1974). When the results of the tests above indicated possible measurement issues (i.e. 1710 profiles spanned among 70*"

*floats), each pre-processing variable time-series was interrupted and only previously-collected profiles were retained (i.e. 9837 stations in BOPAD-prof)."*

[Figure]

*Example of a sharp gradient for b$_{bp}$(700) measurements collected by the float WMO 6901439 operating in the South Atlantic subtropical gyre.*

Page 5 line 21: how did you get the PAR value just below the surface? please provide method? e.g. fitting an exponential
**R38:** PAR just below the surface has been obtained by fitting the profile with a second-degree polynomial function (Organelli et al., 2016a).
**A38:** sentence moved to and expanded in Section 2.2. Please also see **A12**.

Page 6, line 1: was a similar median filter applied to the chla data to remove positive spikes from the chl data set? If so please describe the method used.
**R39:** No median filter was applied on Chl concentrations.
**A39:** Please see **R35/A35**.

Page 6, line 15, figure 3: please include the MLD and Ed on the example profiles
**R40:** The purpose of Section 3 is only to provide context to the database with specific examples. Validation, strengths and limits of the NPQ correction here applied are discussed in Xing et al. (2012). Adding MLD and E$_d$ profiles, with the last ones presented few paragraphs below and in Figure 6, would open discussion to possible biogeochemical vs physical interactions that are beyond the scope of the paper, and which would not be described for the other variables.
**A40:** None.

Page 6. Line 23, figure 3c: is not clear to me how it is possible to retain this shape of profile if the Xing quenching method is applied. unless the MLD is very shallow. In which case what is the Ed relative to the MLD? as it is possible to still have significant quenching below the MLD if the Ed is deeper
**R41:** Yes, the MLD is very shallow (about 7 m). PAR is low (~ 800 µmol quanta m$^{-2}$ s$^{-1}$) and indicates a cloudy day. An unlikely significant NPQ should be found below the MLD.
**A41**: Profile shown in Fig. 3c is retained.

Page 7, line 2: there is no mention in the methods section on quality control about positive spikes being removed from the bbp profiles. On the contrary in the methods section is says that the positive spikes are retained. Please describe the method used to remove positive spikes. Also, please clarify whether the positive spike data set was retained separately so that flux estimates as per Briggs et al., could still be performed?
**R42:** The standard quality control of b$_{bp}$(700) does not remove any positive spike, as outlined in section 2.2. However, for specific studies (e.g., Organelli et al., 2017; Barbieux et al., 2017) positive spikes need to be removed, and this has been described in section 2.4 for BOPAD-surf. Both data versions are shown in Figure 5 and archived in BOPAD-prof. No separate database only containing positive spikes is provided.
**A42:** Page 6, Lines 23-24 (revised ms): in section 2.4 methods for removing positive spikes are now specified clearly: "*A median filter (5 point window) was applied to quality-controlled bbp(700) profiles to identify and subsequently remove positive spikes*"

Page 7, Lines 24-25 (revised ms): "*Both versions of bbp(700) profiles are archived in BOPAD-prof.*" has been added. This is also specified in the section "Data availability" of previous and current versions of the ms.

Page 10, line 10: I am not convinced that you can say anything concrete about the representativeness of the previous model without having a handle on the errors in the bio-argo chla data? if the errors in the chla data can be as much as 100% it is likely that they would significantly affect the shape of the 3 order polynomial fit. In particular all the SO data points which lie above the Morel fit are likely to "in reality" all be shifted to the left (i.e. lower chla) and closer to the Morel model line?

**R43:** The relationship between $Z_{eu}$ vs Chl presented in the first version of the manuscript was intended only as a first guess to establish this relationship using a larger database than the one used in Morel et al. (2007). Possible limitations and critical issues related to application of a factor 2 for resolving sensor calibration were described (Page 10, Lines 12-20, previous ms). However, the effects on the relationship of application of regional calibrations coefficients cannot be evaluated. This is because regional correction factors based on HPLC data and published by Roesler et al. (2017) do not include all the specific areas and periods included in our analysis.

**A43**: To avoid any future misinterpretation of this relationship, we have preferred to remove it from the revised version of the ms. Figure 10 and Table 3 have been modified accordingly.

Page 10, line 20: I think that a really interesting discussion here would be the regional range in errors in chlorophyll associated with the global application of dividing the chla data by 2.

**R44/A44:** please see **R43/A43.**

Page 11, line 1: The discussion does well to highlight the number of profiles, the regional coverage etc but I think what is lacking is a discussion of the benefits of a high resolution long term time series of biological and physical parameters that bio-argo can provide e.g. showing both seasonal and sub-seasonal variability. . . ..and in some cases perhaps even inter-annual variability (depending on the life time of the float or the succession of floats in a similar water mass). I would suggest that this ought to be highlighted with an example time series from one of the floats showing physics (e.g. temp) and biology (e.g. chlorophyll).

**R45/A45:** Page 14, Lines 10-18 (revised ms): the sentence now looks as: "*The two databases presented here can be directly exploited for several applications, from biogeochemistry and primary production estimation and modeling, to the analysis of the physical forcing on biology together with the assessment of any seasonal and sub-seasonal dependence, and to the evaluation of ocean's bio-optical variability. For specific examples based on same PROVOR-CTS4 profiling floats included in this study, the reader is referred to the works by Dall'Olmo and Mork (2014) and Poteau et al. (2017) for estimation and analysis of particulate organic carbon concentrations and fluxes; Lacour et al. (2017), Mignot et al. (2017) and Stanev et al. (2017) for observing physical impacts on biology; and Organelli et al. (2017) and Barbieux et al. (2017b) for analysis of the variability in diffuse attenuation coefficients for downward irradiance and particle optical backscattering-to-chlorophyll ratios across different oceanic areas, respectively. It is worth noting that the latter two studies have been pursued by exclusively exploiting BOPAD-surf and BOPAD-prof.*"*. As the manuscript already contains a high number of figures, no figure showing an example of temperature and chlorophyll time-series has been added.

Page 11, line 15: I think that it would be good to mention some of the other bio-argo data bases that are currently available e.g. SOCCOM and perhaps plans to integrate them if any?

**R46/A46:** Page 15, Line 3 (revised ms): The reference Johnson et al. (JGR, 2017) on the SOCCOM array has been added.

Table 2: Perhaps add the abbreviations to the Basin section of the Table to reflect those in Figure 1.

**R47/A47:** added.

Figure 1. It is hard to see both the surf and prof stations on this Figure. I wonder if they would be clearer if you reduced the size of the blue dots slightly and outlined the red diamonds in black.

**R48/A48:** done. Please see figure below.

[Figure]

*Revised Figure 1*

Please also note the supplement to this comment: https://www.earth-syst-sci-data-discuss.net/essd-2017-58/essd-2017-58-RC2-supplement.pdf Interactive comment on Earth Syst. Sci. Data Discuss., https://doi.org/10.5194/essd-2017-58, 2017.

**R49/A49:** please see **R30**.

**R50/A50:** following modifications suggested by Reviewers, we decided also to:
- Figure 2, plot b: float WMO6901439 replaced by float WMO 6901474. Caption modified accordingly and indicating from which FDOM values data have been discarded.
- remove Table 3 and place statistics of the $K_d(PAR)$ vs $K_d(490)$ relationship in caption of Figure 10.